
# Offline analysis of the chemical composition and hygroscopicity of sub-micrometer aerosol at an Asian outflow receptor site and comparison with online measurements

Yange Deng[1,2,3], Hiroaki Fujinari[1], Hikari Yai[1], Kojiro Shimada[4,5], Yuzo Miyazaki[6], Eri Tachibana[6], Dhananjay K. Deshmukh[7], Kimitaka Kawamura[7], Tomoki Nakayama[2,8], Shiori Tatsuta[4], Mingfu Cai[9,10], Hanbing Xu[9], Fei Li[11,12], Haobo Tan[11], Sho Ohata[13,14,15], Yutaka Kondo[16], Akinori Takami[17], Shiro Hatakeyama[4,18], and Michihiro Mochida[1,2]

[1] Graduate School of Environmental Studies, Nagoya University, Nagoya, 464-8601, Japan
[2] Institute for Space-Earth Environmental Research, Nagoya University, Nagoya, 464-8601, Japan
[3] Now at National Institute for Environmental Studies, Tsukuba, 305-8506, Japan
[4] Faculty of Agriculture, Tokyo University of Agriculture and Technology, Tokyo, 183-8538, Japan
[5] Now at Department of Chemistry, Biology, and Marine Science, University of the Ryukyus, Okinawa, 903-0213, Japan
[6] Institute of Low Temperature Science, Hokkaido University, Sapporo, 060-0819, Japan
[7] Chubu Institute for Advanced Studies, Chubu University, Kasugai, Aichi, 487-8501, Japan
[8] Now at Graduate School of Fisheries and Environmental Sciences, Nagasaki University, Nagasaki, 852-8521, Japan
[9] School of Atmospheric Sciences, Sun Yat-sen University, Zhuhai, Guangdong, 519082, China
[10] Now at Institute for Environmental and Climate Research, Jinan University, Guangzhou, Guangdong, 511443, China
[11] Institute of Tropical and Marine Meteorology, China Meteorological Administration, Guangzhou, 510-640, China
[12] Now at Xiamen Key Laboratory of Straits Meteorology, Xiamen Meteorological Bureau, Xiamen, 361012, China
[13] Department of Earth and Planetary Science, The University of Tokyo, Tokyo, 113-8654, Japan
[14] Now at Institute for Space-Earth Environmental Research, Nagoya University, Nagoya, 464-8601, Japan
[15] Now at Institute for Advanced Research, Nagoya University, Nagoya, 464-8601, Japan
[16] National Institute of Polar Research, Tokyo, 190-8518, Japan
[17] National Institute for Environmental Studies, Tsukuba, 305-8506, Japan
[18] Now at Asia Center for Air Pollution, Japan Environmental Sanitation Center, Niigata, 950-2144, Japan

*Correspondence to*: Michihiro Mochida (mochida@isee.nagoya-u.ac.jp)

**Abstract.** Filter-based offline analysis of atmospheric aerosol hygroscopicity coupled to composition analysis provides information complementary to that obtained from online analysis. However, its application itself and comparison to online analysis have remained limited to date. In this study, daily submicrometer aerosol particles ($PM_{0.95}$, 50 % cutoff diameter: 0.95 µm) were collected onto quartz fiber filters in Okinawa Island, a receptor of East Asian outflow, in the autumn of 2015. The chemical composition of water-soluble matter (WSM) in $PM_{0.95}$ and $PM_{0.95}$ itself, and their respective hygroscopicities were characterized through the offline use of an aerosol mass spectrometer and a hygroscopicity tandem differential mobility analyzer. Thereafter, results were compared with those obtained from online analyses. Sulfate dominated the WSM mass (60 %), followed by water-soluble organic matter (WSOM) (20 %) and ammonium (13 %). WSOM accounted for most (93 %) of the mass of extracted organic matter (EOM) and the atomic O to C ratios (O:C) of WSOM and EOM were high (mean ±



standard deviation were, respectively, 0.84±0.08 and 0.79±0.08), both of which indicate highly aged characteristics of the observed aerosol. The hygroscopic growth curves showed clear hysteresis for most samples. At 85 % RH, the calculated hygroscopicity parameter $\kappa$ of the WSM ($\kappa_{WSM}$), WSOM, EOM, and $PM_{0.95}$ ($\kappa_{PM0.95}$) were, respectively, 0.50±0.03, 0.22±0.12, 0.20±0.11, and 0.47±0.03. An analysis using the thermodynamic E-AIM model shows, on average, that inorganic salts and

WSOM respectively contributed 88 % and 12 % of the $\kappa_{WSM}$ (or $\kappa_{PM0.95}$). High similarities were found between offline and online analysis for chemical compositions that are related to particle hygroscopicity (the mass fractions and O:C of organics, and the degree of neutralization), and also for aerosol hygroscopicity. As possible factors governing the variation of $\kappa_{WSM}$, the influences of WSOM abundance and the neutralization of inorganic salts were assessed. At high RH (70–90 %), the hygroscopicity of WSM and $PM_{0.95}$ was affected considerably by the presence of organic components; at low RH (20–50 %),

the degree of neutralization could be important. This study not only characterized aerosol hygroscopicity at the receptor site of East Asian outflow, but also shows that the offline hygroscopicity analysis is an appropriate method, at least for aerosols of the studied type. The results encourage further applications to other environments and to more in-depth hygroscopicity analysis, in particular for organic fractions.

## 1 Introduction

Hygroscopicity of atmospheric aerosols is a key property related to its effects on climate and air quality. It influences the aerosol's light scattering and absorption ability (Titos et al., 2016; Zhou et al., 2020) and therefore affects visibility and the radiative balance of the Earth. Moreover, it influences the capability of aerosol particles to act as cloud condensation nuclei (CCN) under supersaturated water-vapor conditions, which further influences the radiative balance by affecting the optical property and lifetime of clouds (Mcfiggans et al., 2006). In addition, the absorption of water by aerosol particles might serve

important media for aqueous-phase reactions (McNeill, 2015; Cheng et al., 2016). The hygroscopicity of aerosol particles might also influence their adverse effects on human health: aerosol particle deposition in a human body is expected to depend on hygroscopic growth under high relative humidity (RH) in the respiratory system (Braakhuis et al., 2014).

The hygroscopicity of atmospheric aerosol is governed by the chemical composition. It is often represented by hygroscopicity

parameter $\kappa$. Several hygroscopicity studies have been performed for atmospheric particles or particles generated by extracts from atmospheric aerosol samples. Whereas the ability of the particles to grow to cloud droplet size under supersaturated water vapor conditions has been investigated using a cloud condensation nucleus (CCN) counter, alternatively, the growth of particle size as a result of humidification under sub-saturated conditions has also been investigated, for example using a hygroscopicity tandem differential mobility analyzer (HTDMA). At water activity ($a_w$) of around 0.9 or higher, inorganic salts such as NaCl

and $(NH_4)_2SO_4$ present high $\kappa$ values of 0.5–1.4; atmospheric organic aerosol (OA) components present intermediate $\kappa$ values of 0.01–0.5 (Petters and Kreidenweis, 2007). By contrast, black carbon retains almost no water (Guo et al., 2016) and its $\kappa$ value can be inferred as zero. The $\kappa$ values of ambient aerosol particles are explained by the combination of water uptake by



respective components in the mixture. In the low to middle RH range, the deliquescence and efflorescence of inorganic salts can strongly affect the hygroscopic growth of atmospheric particles, and could result in hysteresis according to the history of RH (Tang et al., 1977). For the hygroscopicity of ambient particles, the composition of inorganics, including the degree of neutralization, affects their contribution to particle hygroscopicity (Tang and Munkelwitz, 1977; Freedman et al., 2019). In

addition, the contrasting hygroscopicity of organics and inorganics are responsible for variations of their hygroscopicity (e.g., Gunthe et al., 2009; Cerully et al., 2011; Pierce et al., 2012; Levin et al., 2014; Deng et al., 2018). The dominant components of atmospheric aerosols govern the dependence of aerosol hygroscopicity on locations: hygroscopicity in the forest atmosphere (Gunthe et al., 2009; Hong et al., 2014), where OA dominates the aerosol composition, is generally less than that in the marine atmosphere (Mochida et al., 2011; Pringle et al., 2010), where inorganic salts dominate. Moreover, while the oxygenation of

OA relates to its hygroscopicity (Kuwata et al., 2013), correlation from analysis of atmospheric aerosols can be poor (Kuang et al., 2020). Whereas multiple compositional factors are expected to control the aerosol hygroscopic growth as explained above, studies elucidating variations of hygroscopic growth under different atmospheric environments are few, which can be attributed to the lack of hygroscopicity analyses coupled with chemical composition analyses.

For characterizing the hygroscopicity of atmospheric aerosols, offline analysis, i.e., the collection of aerosol samples on substrates, followed by analysis of the hygroscopicity of chemical components therein, provides information that complements information obtained from online analysis. Such offline analyses have been conducted for urban aerosols (e.g., Aggarwal et al., 2007; Mihara and Mochida, 2011) and aerosols in remote environments (e.g., Silvergren et al., 2014; Boreddy and Kawamura, 2016). For offline methods, hygroscopicity of aerosol particles with size up to ~1 μm or larger was analyzed,

providing data for hygroscopicity in a wide range of particle sizes, which are often difficult to obtain by online analyses. Furthermore, whereas information related to the mixing state is lost, offline methods enable investigation of the hygroscopicity of specific compound groups in aerosols, for example, water-soluble matter and humic-like substances (Gysel et al., 2004). Moreover, whereas field deployments of online instruments such as HTDMA might be a heavy duty and hinder observations particularly at remote sites, offline analysis can be a good alternative for aerosol hygroscopicity studies. Recent studies have

indicated that offline use of an aerosol mass spectrometer (AMS) can be a useful means to elucidate the contribution of OA component to aerosol hygroscopicity because of its capability of quantifying organic mass in addition to organic carbon, and to characterize the chemical structure of OA (Mihara and Mochida, 2011; Lee et al., 2019). More offline studies, in particular those of the role of OA, should be undertaken to characterize aerosol hygroscopicity further.

Positive and negative artifacts have been evaluated for offline analyses of the concentrations of aerosol chemical components (Turpin et al., 2000; Chow et al., 2005). Sampling artifacts are inherent to offline analyses, and might also affect offline hygroscopic growth measurements. However, the propriety of the offline method for quantifying aerosol hygroscopicity is not evaluated tentatively. Bias might arise from sampling artifacts by adsorption or evaporative loss of compounds and degradation of collected aerosol components, as in the case of the quantification of chemical components. Although full resolution of the


degree of such artifacts is difficult, comparison between offline and online results from measurements of chemical composition and aerosol hygroscopicity is expected to constrain the possible magnitudes of artifacts, and warrant the further utilization of offline methods.

In this study, we analyzed the hygroscopicity of submicrometer (PM$_{0.95}$, 50 % cutoff diameter: 0.95 μm) aerosol samples collected in Okinawa, a remote island in Japan. We interpreted results based on the chemical composition analysis, including the offline use of an aerosol mass spectrometer. Okinawa is considered a receptor site of aerosols from the Asian continent, thereby suited to characterize the nature of the hygroscopicity of aged atmospheric aerosols after long-range transport. Although a few reports have described the relation between the chemical composition and the hygroscopicity in Okinawa

(Mochida et al., 2010; Cai et al., 2017), no report of the relevant literature has described a study of their mutual quantitative relation. Here, based on measurements of the chemical composition of PM$_{0.95}$ and the hygroscopic growth of the extracted water-soluble matter (WSM) in PM$_{0.95}$, the hygroscopicity parameter $\kappa$ of WSM, water-soluble organic matter (WSOM), and PM$_{0.95}$ at 20–90 % RH are characterized. Factors responsible for the hygroscopicity parameters are assessed. This study, an extension of our online aerosol hygroscopicity study (Cai et al., 2017), aims at characterizing the RH and composition

dependence of the hygroscopicity of aged aerosols after their atmospheric transport. In addition, from a methodological viewpoint, the offline analysis of the composition and hygroscopic growth using filter samples are assessed by comparison with online analysis.

## 2 Experimental

### 2.1 Aerosol sampling and extraction

Aerosol samples were collected at Cape Hedo Atmosphere and Aerosol Monitoring Station (128.15 °E, 26.52 °N) of the National Institute for Environmental Studies, Japan, in Okinawa Island during 26 October and 9 November, 2015. It is a receptor site of Asian outflow after long-range transport (Takami et al., 2007; Lun et al., 2014). The aerosols were collected daily on quartz fiber filters using a high-volume aerosol sampler (Model-120 B; Kimoto Electric Co. Ltd.) equipped with a cascade impactor (TE-234; Tisch Environmental Inc.). Details of sampling periods for the respective samples are presented in

Table S1. The respective means of the RH, air temperature, and wind speed during the sampling days were 75.4 %, 23.8 °C, and 3.8 m s$^{-1}$. Precipitation was only observed on 30 October (Fig. S1). The PM$_{0.95}$ samples collected on backup filters were used for analysis. The quartz fiber filters were pre-combusted at 450 °C for 6 h before use. The high-volume sampler was placed on the rooftop of a station building. Its inlet was located about 4 m above the ground. The flow rate of the sampler was about 1,100 L min$^{-1}$. For each sample, about 1,600 m$^3$ of air was aspirated. Blank samples were collected by operating the

sampler for only 10 s. After sample collection, the filter samples were stored in freezers before analysis.



For offline analyses using the HTDMA and the AMS, WSM and water-insoluble organic matter (WISOM) in each aerosol sample were extracted as follows. First, three punches (34 mm diameter) from each filter sample were ultrasonicated with 3 g water for 15 min. The solution was then filtered with a Teflon filter (0.20 μm, Millex-FG; Millipore Corp.). For each aerosol sample, the extraction was repeated three times and the WSM solutions were combined. Then, the WISOM in the same sample

punches were extracted by ultrasonication with first 3 g of methanol once and then 3 g of dichloromethane/methanol (2/1, v/v) mixture three times. The extract solutions were filtered through the Teflon filter used for the filtration of WSM, and the solutions after filtration were combined. The combined WISOM solution was dried with a rotatory evaporator and was re-dissolved in dichloromethane/methanol (2/1, v/v) solution. For TOC analyses for WSM, three punches (diameter: 34 mm) of each filter sample were extracted ultrasonically once with 20 ml ultrapure water for 15 min before being filtered through a

syringe filter; similarly, for IC analyses for WSM, one punch with a diameter of 34 mm was extracted with 10 ml ultrapure water (20 min ultra-sonication) and then filtered (Müller et al., 2017b).

## 2.2 Hygroscopic growth measurement for WSM

The hygroscopic growth factors ($g_f$) of WSM at 20, 30, 40, 50, 60, 65, 70, 75, 80, 85, and 90% RH in humidification and dehumidification branches were obtained using an HTDMA. For measurements, a WSM solution was nebulized using a home-

made nebulizer equipped with a syringe pump to generate WSM aerosol particles. After the generated WSM aerosol was passed through a Nafion humidifier (NH1, MH-110-12F-4; Perma Pure LLC), it was dried with two diffusion dryers in series containing silica gel (White, Middle Granule; Kanto Kagaku) and a molecular sieve (13X/4A mixture; Supelco and Sigma-Aldrich). The dried aerosol flow was transferred through an impactor (model 1035900; TSI Inc.) with a 0.071 cm diameter orifice in the front. It was then neutralized using an $Am^{241}$ neutralizer. The neutralized aerosol was passed through the first

differential mobility analyzer (DMA1, Model 3081; TSI Inc.) in the HTDMA. The aerosol particles with 100 nm dry diameter were selected. In humidification mode, dry 100 nm aerosol particles were then humidified using a second Nafion humidifier (NH2, MD-110-24S-4; Perma Pure LLC). In dehumidification mode, dry 100 nm particles were first humidified to > 97 % RH using a third Nafion humidifier (NH3, MD-110-24S-4; Perma Pure LLC) before the particles were transferred to NH2.

The aerosol particles downstream of NH2 were scanned using a second DMA (DMA2, Model 3081; TSI Inc.) coupled with a condensation particle counter (CPC, model 3775; TSI Inc.). The aerosol flow rates of both DMA1 and DMA2 were 0.3 L min$^{-1}$. The RH values at the outlet of sheath flow of the DMA1, the inlet of NH2, the inlets of sample and sheath flows of the DMA2, and the outlet of sheath flow of the DMA2 were monitored using RH sensors (HMT337; Vaisala). During the experiment, the RH inside the DMA1 was lower than 10 %. The residence time from the outlet of the NH2 to the inlet of

DMA2 was calculated as 13 s. The $g_f$ at 90 % RH in humidification and dehumidification branches was measured separately one month later than the $g_f$ at other RH. The $g_f$ is defined as the ratio of the mobility diameter of particles classified using DMA2 to the dry mobility diameter (100 nm), which were retrieved using the Twomey algorithm as described by Mochida et al. (2010). The mode $g_f$ of fitted lognormal distributions under different RH conditions were used to represent the hygroscopic





growth of WSM and for the derivation of the hygroscopicity parameter ($\kappa_{WSM}$) following the $\kappa$- Köhler theory (Petters and Kreidenweis, 2007) as

$$\kappa_{WSM} = (g_f^3 - 1)\left[\frac{\exp\left(\frac{4\sigma M_w}{RT\rho_w d_{wet}}\right)}{RH} - 1\right], \tag{1}$$

where $\sigma$ represents the surface tension at the solution–air interface, $M_w$ and $\rho_w$ respectively denote the molecular mass and

density of pure water, $d_{wet}$ stands for the product of $g_f$ and $d_{dry}$ (here is 100 nm), $R$ is the universal gas constant, and $T$ is the absolute temperature. In Eq. (1), $T$ of 298.15 K was used considering the temperature at the outlet of sheath flow of DMA2 (24.22–26.59 °C). The surface tension of pure water was $\sigma$. The equations used to calculate the density and surface tension of pure water and the densities of dry inorganic salts are the same as those used in the online Extended AIM Aerosol Thermodynamics (E-AIM) model (Sect. 2.5). Measurement data were used to derive $g_f$ and $\kappa_{WSM}$ only if the RH values at the

outlet of DMA2 meet certain criteria (Text S1).

Before hygroscopic growth measurements for aerosol extracts, the size selection performance of DMA1 and DMA2 was assessed using PSL particles of standard size (Text S2). Furthermore, the hygroscopic growth of pure ammonium sulfate (AS, 99.999 % trace metals basis; Aldrich) particles were measured following the same procedure as that for the WSM samples to

confirm the HTDMA performance (Text S3). The $g_f$ of dry AS particles (RH = 7.22±0.04 %) was measured to quantify the slight difference of sizing (1.9 %) between the two DMAs. This difference has been corrected for derivation of $g_f$ of WSM samples and AS particles. More details about the quality control of the offline analyses are presented in Text S4.

### 2.3 Chemical composition analyses

Ammonium, nitrate, sulfate, sodium, potassium, calcium, magnesium, chloride, and methane sulfonic acid (MSA) in WSM

samples were quantified using an ion chromatograph (Model 761 compact IC; Metrohm AG). Concentrations of water-soluble organic carbon (WSOC) in WSM samples were determined using a total organic carbon analyzer (Model TOC-L$_{CHP}$; Shimadzu Corp.). The results are presented in Table S3.

To characterize the chemical structures of WSOM and WISOM and to quantify their concentrations, WSM and WISOM

samples were analyzed using a high-resolution time-of-flight mass spectrometer (AMS; Aerodyne Research Inc.; Decarlo et al., 2006) by nebulizing the solutions using Ar and by transferring the generated particles to the AMS. Before analysis by AMS, the WSM aerosol flow was dried using two diffusion driers filled with silica gel. The WISOM aerosol flow was dried by two diffusion driers filled in series with activated carbon (to remove dichloromethane and methanol vapor) and silica gel. The AMS were operated in both V-mode and W-mode. The W-mode data were analyzed to obtain the atomic ratios of O to C

(O:C), H to C (H:C), and organic mass to organic carbon (OM:OC) based on the Improved-Ambient method (Canagaratna et





al., 2015) for WSOM and WISOM. The O:C and H:C of WSOM and WISOM were used further to derive their densities (Kuwata et al., 2012). The mass concentration of WSOM was calculated as the product of WSOC from the TOC analyzer and the OM:OC of WSOM. To validate the quantification of WSOM and derive the mass ratios of WISOM to WSOM, the mass spectra of WSOM and WISOM, and those of the mixtures of phthalic acid and WSOM (or WISOM) from V-mode AMS

analysis were used for their quantification (Text S5). The mass concentration of WISOM was calculated as the product of the mass concentration of WSOM and the mass ratio of WISOM to WSOM from the mass spectral analysis. The mass concentration of extracted organic matter (EOM) was defined as the sum of WSOM and WISOM. The mass concentration of water-insoluble organic carbon (WISOC) was derived by dividing the mass concentration of WISOM by the OM:OC of WISOM. The extracted organic carbon (EOC) was defined as the sum of WSOC and WISOC.

The concentrations of OC and elemental carbon (EC) in $PM_{0.95}$ aerosol samples were found using a Sunset Laboratory carbon analyzer with the Interagency Monitoring of Protected Visual Environments (IMPROVE_A) temperature protocol and the thermal-optical transmission method. A filter punch of 16 mm diameter was used for the analysis, and the presence of carbonate carbon was not considered. Good agreement (Fig. S5) was found between EOC and OC, indicating high recovery of EOM.

### 2.4 Concurrent online measurements of ambient aerosol

During the period of the filter sampling of $PM_{0.95}$, the mass concentrations of non-refractory chemical components (sulfate, nitrate, ammonium, chloride, and organics) and black carbon (BC) in $PM_1$ (50 % cutoff diameter: 1 μm) were measured respectively using the same AMS as that for the offline analysis, and a filter-based absorption photometer continuous soot

monitoring system (COSMOS; Kanomax, Osaka, Japan) (Mori et al., 2014; Ohata et al., 2019). Furthermore, the number–size distributions of submicrometer aerosols were measured using a scanning mobility particle sizer (SMPS) composed of a DMA (model 3081; TSI Inc.) and a water-based CPC (model 3785; TSI Inc.). The AMS was operated in both V + pToF-mode and W-mode with time resolution of 30 min. The bulk mass concentrations of non-refractory aerosol components were derived from V-mode data. Composition-dependent collection efficiency (Middlebrook et al., 2012) was applied for quantification.

The W-mode data were analyzed to obtain the O:C and H:C and densities of organics in the manner of the offline analysis. The SMPS measured the aerosol number-size distributions at diameters of 13.8–749.9 nm every 5 min. The DMA in the SMPS was operated with an aerosol flow rate of 0.3 LPM and a sheath to aerosol flow ratio of 10:1. Compressed dry pure air was supplied to the CPC through an equalizer to complement its total inlet flow rate of 1.0 LPM. Temperature and RH of ambient air, wind speed and direction, and precipitation were measured using a weather sensor (model WXT520; Vaisala). The AMS

was calibrated before both online and offline (Sect. 2.3) measurements using the same procedures as those reported by Deng et al. (2018). The SMPS was calibrated using standard size PSL particles (Text S2) before ambient measurements. Furthermore, a hygroscopicity and volatility tandem differential mobility analyzer (H/V-TDMA) was deployed during 1–9 November 2015 to measure the size-resolved aerosol hygroscopicity and volatility. Related details have been presented by Cai et al. (2017).



For comparison between offline and online data, the time windows for offline data were truncated to 10 am to 10 am (24 h). Online data were averaged for the one-day periods (Table S1).

**2.5 Prediction of WSM hygroscopicity based on E-AIM model**

Hygroscopic growth of the WSM sample for the water activity ($a_w$) range of 0.10–0.99 was predicted without considering the water uptake by WSOM using the online Extended AIM Aerosol Thermodynamics Model III (E-AIM III, http://www.aim.env.uea.ac.uk/aim/model3/model3a.php, last access: 1 August 2019; Clegg et al., 1998; Wexler and Clegg, 2002). The inorganic chemical components of WSM (sulfate, sodium, and ammonium) obtained from IC analysis and the WSOM obtained from TOC and offline AMS analyses were used for derivation. Potassium, calcium, magnesium, nitrate, and chloride were not considered in the E-AIM because of their very low concentrations (Table S3). The RH-dependent hygroscopicity parameters of WSM, $\kappa_{WSM}$, were predicted from hygroscopic growth data following the $\kappa$- Köhler theory. The RH-dependent hygroscopicity parameters of water-soluble inorganic matter (WSIM) in each WSM sample, $\kappa_{inorg}$, were derived similarly to those for $\kappa_{WSM}$. Details of these derivations are presented in Text S6.

**2.6 Estimating the hygroscopicity of WSOM, EOM, and PM$_{0.95}$**

The hygroscopicity parameters of WSOM ($\kappa_{WSOM}$), EOM ($\kappa_{EOM}$), and PM$_{0.95}$ ($\kappa_{PM0.95}$) were calculated on the assumption that the volumes of water retained by respective components are additive (Petters and Kreidenweis, 2007):

$$\kappa_{WSM} = \varepsilon_{WSOM/WSM}\kappa_{WSOM} + \varepsilon_{WSIM/WSM}\kappa_{inorg} \tag{2}$$

Therein, $\kappa_{WSM}$ is the hygroscopicity parameter of WSM particles; $\kappa_{WSOM}$ and $\kappa_{inorg}$ respectively denote hygroscopicity parameters of WSOM and WSIM. The $\varepsilon_{WSOM/WSM}$ and $\varepsilon_{WSIM/WSM}$ respectively stand for the volume fractions of WSOM and WSIM in WSM, as derived from offline IC, TOC, and AMS analyses (Text S7).

The hygroscopicity parameter of EOM was estimated on the assumption that the hygroscopicity parameter of WISOM, $\kappa_{WISOM}$, is zero, as

$$\kappa_{EOM} = \varepsilon_{WSOM/EOM}\kappa_{WSOM} + \varepsilon_{WISOM/EOM}\kappa_{WISOM} = \varepsilon_{WSOM/EOM}\kappa_{WSOM}, \tag{3}$$

where $\varepsilon_{WSOM/EOM}$ and $\varepsilon_{WISOM/EOM}$ respectively represent the volume fractions of WSOM and WISOM in EOM (Text S7). With consideration of EC and WISOM but neglecting other water-insoluble inorganics in PM$_{0.95}$, the hygroscopicity parameter of PM$_{0.95}$ was also estimated.

$$\kappa_{PM0.95} = \varepsilon_{WSM/PM0.95}\kappa_{WSM} + \varepsilon_{WISOM/PM0.95}\kappa_{WISOM} + \varepsilon_{EC/PM0.95}\kappa_{EC}$$

$$= \varepsilon_{WSM/PM0.95}\kappa_{WSM} \tag{4}$$


Therein, $\varepsilon_{\mathrm{WSM/PM0.95}}$, $\varepsilon_{\mathrm{WISOM/PM0.95}}$, and $\varepsilon_{\mathrm{EC/PM0.95}}$ respectively represent the volume fractions of WSM, WISOM, and EC among the sum of these three components (Text S7). Here, the hygroscopicity of EC, $\kappa_{\mathrm{EC}}$, was assumed to be zero.

### 3 Results and Discussion

#### 3.1 Mass concentrations and composition of aerosol components

The atmospheric mass concentrations of chemical components in $PM_{0.95}$ samples and their mass fractions are presented in Fig. 1 along with the mass concentrations of chemical components and number-size distributions of aerosols from online analyses. Offline analysis of $PM_{0.95}$ samples indicated that sulfate was most abundant (mean ± standard deviation: 2.62±1.70 µg m$^{-3}$) throughout the observation period, followed by WSOM (0.86±0.51 µg m$^{-3}$), ammonium (0.59±0.32 µg m$^{-3}$), EC (0.10±0.03 µg m$^{-3}$), WISOM (0.10±0.14 µg m$^{-3}$), and sodium (0.07±0.03 µg m$^{-3}$). Accordingly, sulfate accounted for the largest mass

fraction (59.5 %) among the quantified $PM_{0.95}$ components, followed by WSOM (19.7 %), ammonium (13.4 %), EC (2.36 %), WISOM (2.33 %), and sodium (1.50 %). The contributions of potassium, magnesium, calcium, nitrate, and chloride to the $PM_{0.95}$ samples were small: 1.23 % in total. WSOM accounted for a major fraction of EOM (mean of 93 % on a mass basis). This large proportion suggests that the studied aerosol was aged substantially, considering the much lower proportions against total OM in East Asian suburban (approx. 60 %; Müller et al., 2017a) and urban environments (27–45 %; Miyazaki et al.,

2006), which are based on our mass conversions, assuming factors of 1.8 and 1.2 to convert WSOC to WSOM and WISOC to WISOM, respectively. The mass ratio of EC to EOM was on average 11 %, which is similar to the proportion to total OM (12 %) based on earlier reported OC:EC over the Sea of Japan and offshore of Japan (Lim et al., 2003; a factor of 2.1 (Turpin and Lim, 2001) was assumed to convert OC to OM). As shown in Fig. 1d, the aerosol number–size distribution shows bimodal or broad unimodal characteristics.

For sulfate, organics, ammonium, and EC (BC), the relative abundances among them from the offline analysis showed moderate agreement with those from the online analysis (61.9 %, 22.0 %, 13.8 %, and 2.4 %, respectively, for sulfate, EOM, ammonium, and EC from offline measurements during the period with effective data). The coefficients of determination ($r^2$) of the mass fractions of sulfate, organics, ammonium, and EC (BC) in those four aerosol components from offline and online

analyses were, respectively, 0.62, 0.31, 0.22, and 0.09 (Fig. S6). Whereas the absolute concentrations of the aerosol are not crucially important for the offline analysis of aerosol hygroscopicity, high positive correlations between online and offline measurements for sulfate, organics, ammonium, and EC (BC) ($r^2$ of which are, respectively, 0.90, 0.83, 0.92, and 0.76; Fig. S6) support agreement between offline and online analyses. The average mass concentrations of sulfate, organics, and ammonium from online measurements were, respectively, 77 %, 77 %, and 71 % of offline results. Lower concentrations

might result from uncertainty of the collection efficiency of the online AMS analysis (Takegawa et al., 2009), and from different size windows for offline $PM_{0.95}$ sampling and online AMS analysis. Sampling bias of organics or ammonium by absorption/evaporation does not explain the difference because sulfate measurements should not be influenced largely from



positive/negative artifacts, considering its low volatility (Johnson et al., 2004) and reported absence of $SO_2$ artifacts for filters of other types (Eldred and Cahill, 1997). The mean mass concentration of BC from online analysis was almost equal to those of EC from the offline analysis: the ratio of the former to the latter was 1.02.

Figure 2 presents backward air mass trajectories for $PM_{0.95}$ sampling. For most days, the three-day trajectories passed over the Asian continent and/or the Japan archipelago, but maritime air masses also arrived at the observation site during 6–8 November 2015. A comparison between air mass trajectories and aerosol concentration data shows that maritime air masses during 6–8 November are characterized by lower aerosol mass concentrations, but higher mass fractions of sodium ($\geq 5$ %) than the other days influenced by continental air masses. The mean mass concentrations of sulfate, WSOM, ammonium, and EC from offline

analyses during 6–8 November were, on average, 1/6, 1/5, 1/5, and 1/2 of those during other days, respectively, whereas the mean mass concentration of sodium during the period (0.07 µg m$^{-3}$) was similar to that of other days (0.06 µg m$^{-3}$).

Two types of composition related to aerosol hygroscopicity were investigated: the O:C of organics and the molar ratio of ammonium to sulfate (after omitting sulfate that was preferentially neutralized by sodium) ($R_{A/S}$; Text S7), which represents

the degree of neutralization of sulfate by ammonium and sodium. For the derivation of $R_{A/S}$, the neutralization of sulfate by other cations was not considered because their contributions were small. The O:C of EOM from $PM_{0.95}$ samples is presented in Figure 3a. The mean ± standard deviation of the ratio was 0.79±0.08. Although the value was 16 % lower than that of the mean O:C of OA from the online analysis of $PM_1$ (0.95±0.09), they were in good agreement ($r^2$=0.58; Fig. S7). The O:C values of EOM were in the range of 0.64–0.94, suggesting a highly aged nature of the observed OA (Canagaratna et al., 2015). The

$R_{A/S}$ values from $PM_{0.95}$ samples are presented in Fig. 3b: the mean ± standard deviation was 1.45±0.34. The results suggest that except for the aerosols on 7 and 8 November, the studied aerosols were fairly acidic. To compare the offline and online analyses, $R_{A/S}$ was also derived by ignoring the neutralization of sulfate by sodium, which is presented as $R_{A/S}'$ (Fig. 3b). The mean ± standard deviation of $R_{A/S}'$ from the offline analysis (1.29±0.21) was similar to that from the online analysis (1.29±0.40). The two also showed good agreement ($r^2$=0.52; Fig. S7). If a portion of sulfate was in the form of sodium sulfate at the time

of online AMS analysis, then this fraction might have not been detected considering the high melting temperature of the salt. However, the offline analysis suggests that the fraction of sulfate neutralized by sodium was, on average, only 5 %. Hence, it is not expected to affect the comparison strongly.

On 7 and 8 November, the days under the influence of maritime air masses, the O:C and $R_{A/S}$ from the offline analysis values

were, respectively, lower and higher than those during other periods when the air masses were from the Asian continent and/or the Japan archipelago (Figs. 2 and 3). In addition, comparison between $R_{A/S}'$ and $R_{A/S}$ from the offline analysis shows that sodium neutralized a larger fraction of sulfate on the two days. The results suggest that air masses from the Asian continent transported more aged and acidic aerosol, and that air masses from the North Pacific included less-oxygenated and more-neutralized aerosol. However, it is noteworthy that the possible influence of the external mixing state on the neutralization of



aerosols is not considered. The more-acidic nature of the continental aerosol is expected to be contributed by the formation of sulfate during the transport. In addition, the oxidation of MSA from marine biological activity is expected to contribute to sulfate. The low relative abundance of sodium in the continental aerosol also accounted for the more-acidic nature.

## 3.2 Hygroscopicity of WSM and PM$_{0.95}$

The mean ± standard deviation of the measured $g_f$ values for WSM particles are presented in Fig. 4 and Table S4. The mean $g_f$ values predicted from the E-AIM model without consideration of the water retained by WSOM are also shown in the figure. The $g_f$ values of the respective WSM samples are presented in Fig. S8. The mean ± standard deviation of $g_f$ at 40, 60 and 85 % RH in the humidification (dehumidification) branch were, respectively, 1.04±0.02 (1.09±0.03), 1.13±0.05 (1.22±0.01) and 1.53±0.03 (1.53±0.02) (Table S4). The obtained $g_f$ of WSM at 90 % RH ($g_f(90\%)$) was slightly lower than that of the WSM from Chichijima Island (1.76–1.79), which was also influenced by transport from East Asia but was much farther to the east of the Asian continent compared with Okinawa (Boreddy et al., 2014; Boreddy and Kawamura, 2016). It was also lower than the mean values for WSM during a cruise over the East China Sea (1.99, Yan et al., 2017), which was nearer the Asian continent. On the other hand, the obtained $g_f(90\%)$ of WSM was higher than that of the WSM obtained during a cruise over the Bay of Bengal (1.25–1.43, Boreddy et al., 2016), which was influenced by anthropogenic or biomass burning air masses. For three studies (Boreddy et al., 2014; Boreddy and Kawamura, 2016; Yan et al., 2017), the WSM were extracted from total suspended particles that contain higher mass fractions of inorganics and sea salts than those examined for this study. By contrast, the WSM in the last referred study (Boreddy et al., 2016) was extracted from PM$_{2.5}$ (50 % cutoff diameter: 2.5 μm) with higher mass fractions of organics. These compositional differences should explain the observed differences of $g_f(90\%)$.

Hysteresis of the hygroscopic growth of the WSM particles was observed for most samples except for those collected on 26 October and 2 and 6 November (Fig. S8). The hysteresis was expected to have been caused by the influence of inorganic salts, as indicated by the differences in the predicted hygroscopic growth in humidification and dehumidification branches from the E-AIM model, where only the water retained by inorganics is considered. Being different from the observation, the hysteresis was predicted for almost all samples, which might result from the uncertainty in the quantification of inorganic salts and/or the influence of organic components on the hygroscopicity of WSM (Choi and Chan, 2002). The deliquescence of WSM in the humidification branch was observed in the RH of 50–70 %. In this branch, the WSM shows prominent water uptake at RH as low as 20 % (Fig. 4a), being in contrast to the absence of hygroscopic growth of pure AS (Fig. S2). Water uptake of WSM at low RH in the humidification branch can be enhanced by highly acidic conditions (Sect. 3.1) and/or the presence of WSOM (Gysel et al., 2014). In the dehumidification branch, efflorescence was not evident down to 30 % RH for most samples, indicating the existence of metastable conditions to retain water after experiencing high RH. The samples collected on 7 and 8 November, all characterized by a large sodium fraction, showed clearer efflorescence behavior at 40 % RH (Fig. S8). The high efflorescence RH (ERH) of sodium sulfate (57–59%) (Tang, 1996) might have been associated with the observed


efflorescence. In addition, the high $R_{A/S}$ (2.01 and 1.97 respectively) on these two days could have contributed to the high ERH, which is supported by the distinctive ERH among different forms of ammoniated sulfate (Tang and Munkelwitz, 1994). Whereas the external mixing state of atmospheric aerosol is lost by filter sampling, the former possibility implies that the sea-salt component enhances the ability to effloresce once mixed with other inorganic components. This characteristic, however, is expected to be important only if such aerosols are transported to drier environments.

The obtained $g_f$ values as a function of RH were converted to corresponding $\kappa$ values. The mean ± standard deviation of the measured $\kappa$ values for WSM as a function of RH are presented in Fig. 4b and Table S4. The $\kappa$ values from the E-AIM by ignoring the water uptake by organics are also shown in the same figure. The $\kappa_{WSM}$ of respective WSM samples are presented in Fig. S9. In the humidification branch, the measured $\kappa_{WSM}$ averaged for each RH were 0.17–0.24 (at 20–50 % RH) and 0.50–0.56 (at 70–90 % RH), respectively below and above the marked increase in $\kappa_{WSM}$ with the increase in RH, presumably indicating the deliquescence of major inorganic salts. Comparison between measured $\kappa_{WSM}$ versus predicted $\kappa_{WSM}$ shows that, on average, the measured $\kappa_{WSM}$ were greater than predictions for all RH, suggesting the ubiquitous contributions of WSOM to the measured $\kappa_{WSM}$. The results at RH < 70 % were more deviated from the 1:1 line than those at RH ≥ 70 % RH (Fig. S10), which might indicate dominant contributions of WSOM to $\kappa_{WSM}$ at low RH for some aerosol samples (Gysel et al., 2004; Aggawarl et al., 2007).

In the dehumidification branch, except for the case at 20 % RH, where the corresponding $\kappa$ value was 0.17, the $\kappa$ values of WSM were modestly high, with values of 0.42–0.57. The lack of a large dependence on RH suggests that efflorescence did not occur. Even if it did, it was for minor fractions of inorganics. The contribution of WSOM to the hygroscopicity of WSM was evident from the fact that, except the sample collected on 1 November, the E-AIM model predicted $\kappa$ by ignoring the water retained by WSOM at RH ≥ 65 % were lower than the measured values (Fig. S9). The $\kappa$ values of WSM from respective PM$_{0.95}$ samples in dehumidification branches are presented in Fig. 4c. At high RH (≥ 65 %), the difference in $\kappa_{WSM}$ among different samples was small compared with that at low RH, indicating that the difference in the composition among aerosol samples did not result in large variation in the hygroscopicity of WSM at these RH conditions. Clear variations in hygroscopicity among samples at low RH can be explained by the influence of the degree of neutralization of inorganic salts and the abundance of organics. For example, the $\kappa_{WSM}$ on 1 November at ≤ 60 % RH was higher than on other days, which was likely to be related to a low $R_{A/S}$ ratio (approx. 0.80), as evidenced by the large E-AIM predicted $\kappa_{WSM}$ on this day (Fig. S9); the $\kappa_{WSM}$ on 26 October was lower than that on other days, which might be explained by the high mass fraction of WSOM (Fig. 1b) in addition to the high $R_{A/S}$ (1.66). The contribution of chemical composition will be discussed further in later sections.

By considering the atmospheric concentrations of WISOM and EC, the $\kappa$ values of PM$_{0.95}$ were estimated (Table S4). In the humidification branch, the $\kappa_{PM0.95}$ were 0.15–0.22 and 0.47–0.53, respectively, without (20–50 % RH) and with (70–90 % RH)





the deliquescence of WSM particles (Table S4). In the dehumidification branch, except for the case at 20 % RH, where the corresponding $\kappa_{PM0.95}$ was 0.16, they were in the range of 0.40–0.54 (Table S4). At 90 % RH, $\kappa_{PM0.95}$ was in the range of 0.47–0.52, which is higher than that measured at a supersite in Hong Kong (0.18–0.48, Cheung et al., 2015) that was influenced by clean maritime air masses and/or polluted Asian continental and coastal inflows.

Figure 5 presents a comparison of calculated $\kappa_{PM0.95}$ at 85 % RH with that for 40–200 nm particles from the online analysis ($\kappa_{online}$) at 85 % RH during the same observation period. The estimated range of $\kappa_{PM0.95}$ at 85 % RH (0.49±0.02) was within the range of $\kappa_{online}$ (0.44–0.51) for ambient aerosol particles. The mean number/volume concentrations as a function of dry particle diameter are also depicted in Fig. 5. The clear bimodal shape of the mean aerosol number–size distribution with Aitken and accumulation modes suggests that the aerosols experienced in-cloud processing (Mochida et al., 2011; Hoppel et al., 1986). Similar high hygroscopicity of particles in the Aitken mode (40 nm diameter) and the accumulation mode (150 and 200 nm diameters) suggests the dominance of sulfate in both modes (Mochida et al., 2011). The mean aerosol volume–number concentration presented unimodal distribution with mode diameter of 260 nm, indicating that aerosol mass in the accumulation mode dominates the total aerosol mass in the submicrometer size range. With regard to the influence of aerosol hygroscopicity on aqueous-phase chemical reactions on mass basis, the hygroscopicity of large aerosol particles might be more important. Offline analysis extended the online hygroscopic analysis of <200 nm particles to the whole submicrometer size range, where most of the aerosol liquid water mass exists. The $\kappa_{PM0.95}$ and $\kappa_{online}$ at 85 % RH for 200 nm particles are compared for the days when both data are available (Fig. 6), which showed moderate positive correlations: $r^2$ of 0.15 and 0.31 respectively for dehumidification and humidification branches. Results obtained from comparison of $\kappa_{PM0.95}$ and $\kappa_{online}$ indicate that offline aerosol hygroscopicity analysis can be used as an alternative method, at least for the studied type of aerosols, for which the sampling bias for semi-volatile ammonium nitrate is not significant because of its low abundance.

### 3.3 Hygroscopicity of WSOM and EOM

The hygroscopicity parameters of WSOM and EOM were calculated based on $\kappa_{WSM}$ from measurements in the dehumidification branch and the predicted water uptake by inorganic salts (Eqs. 2 and 3). The results are presented in Fig. 7 and Table S4. At 75–85 % RH, where the deviation of the measured $\kappa$ of AS from that predicted from the E-AIM model was slight ($\leq 5$ %; Fig. S2), $\kappa_{WSOM}$ values were 0.19–0.22. Those values were higher than the $\kappa$ of WSOM from US national parks and Storm Peak Laboratory (0.05–0.15 at RH = 90 %; Taylor et al., 2017) and from fresh Indonesian peat burning particles (0.18 at RH = 85 %; Chen et al., 2017). In the same RH range, $\kappa_{EOM}$ was in the range of 0.17–0.20, and was, on average 9 % lower than that of $\kappa_{WSOM}$. The $\kappa_{EOM}$ from this study was higher than the $\kappa$ of OA in the Western/Central Los Angeles Basin that was influenced by marine air masses (0.14 at RH = 74–92 %; Hersey et al., 2011). It was also higher than the $\kappa$ of OA that were influenced by marine air masses over the continental United States, Canada, the Pacific Ocean, and the Gulf of Mexico and were aged (O:C = 0.93 ± 0.30) (0.13 at RH = 70–95 %; Shingler et al., 2016). However, it might be lower than the $\kappa$ of



OA in a supersite in Hong Kong, for which only an upper limit value of 0.29 (at 90 % RH) was reported (Yeung et al., 2014). It is noteworthy that the estimated mean $\kappa_{WSOM}$ and $\kappa_{EOM}$ of approx. 0.2 is higher than the default $\kappa$ value of organics ($\kappa_{org}$) of 0.14 used in an atmospheric aerosol model (Kawecki and Steiner, 2018). The different $\kappa_{org}$ values of different types or different atmospheric regions reported in this study and earlier studies described above suggest the importance to consider different $\kappa_{org}$

values depending on the types and origins of OA in model calculations. The $\kappa_{WSOM}$ and $\kappa_{EOM}$ values derived from the measurement in the humidification branch at 85 % RH where WSM particles would be mostly or fully dissolved in water were also presented in Table S4. The mean values in the humidification branch were slightly higher than those in the dehumidification branch, but the characteristics explained above also apply to this condition. The fractional contributions of WSOM to the water uptake by WSM and PM$_{0.95}$, represented respectively as $(\varepsilon_{WSOM/WSM} \times \kappa_{WSOM})/\kappa_{WSM}$ and

$(\varepsilon_{WSOM/PM0.95} \times \kappa_{WSOM})/\kappa_{PM0.95}$, are presented in Table S5. The contribution of WSOM to the water uptake by WSM and PM$_{0.95}$ was 10–12 % at 75–85 % RH.

The high $\kappa_{WSOM}$ and $\kappa_{EOM}$ values are reasonably explained by the high O:C ratios of WSOM and EOM on a mean basis. The mean O:C of WSOM and EOM were 0.84 and 0.79, from which $\kappa_{WSOM}$ and $\kappa_{EOM}$ were estimated respectively as 0.19 and 0.18

based on the reported regression lines between $\kappa_{org}$ and O:C (Chang et al., 2010; Lambe et al., 2011; Wu et al., 2013; Deng et al., 2018) (Fig. 7). Correlations of $\kappa_{WSOM}$ and $\kappa_{EOM}$ with the O:C were weak (Fig. S11, dehumidification branch). That weakness, however, should not contradict earlier reported positive correlation between $\kappa_{org}$ and O:C, given the narrow range of O:C observed in this study (0.64–0.94). The absence of correlation might also be related to the fact that the O:C values of WSOM and EOM fall in the plateau in the high O:C range reported by Cappa et al. (2011), who reported sigmoidal dependence of the

hygroscopicity of OA on the O:C.

### 3.4 Factors affecting the hygroscopicity of WSM and PM$_{0.95}$

The discussion presented in earlier sections indicates that the water uptake of WSOM and the degree of neutralization of the inorganic components influence the hygroscopicity of WSM. Here, the influences of WSOM and $R_{A/S}$ on $\kappa_{WSM}$ and $\kappa_{PM0.95}$ at

20–90 % RH are assessed in light of the variations of the hygroscopicity.

The relation between the mass fraction of WSOM ($f_{WSOM}$) and $\kappa_{WSM}$ at 20–90 % RH in the dehumidification branch is presented in Figs. 8a and 8b. Despite the narrow range of $f_{WSOM}$, moderate negative correlation between $f_{WSOM}$ and $\kappa_{WSM}$ was observed for all RH conditions, except for 90 % RH, indicating the importance of the relative contributions of WSOM and WSIM

(mainly sulfate + ammonium) to the hygroscopicity of WSM. The poor correlation between $f_{WSOM}$ and $\kappa_{WSM}$ at 90 % RH in the dehumidification branch was probably attributable to measurement uncertainty. Comparison between $f_{WSOM}$ and $\kappa_{WSM}$ in the humidification branch shows high correlation, as presented in Fig. 8b. This dependence is explained by the low



hygroscopicity of WSOM compared to that of WSIM. The shaded areas in Fig. 8b represent $\kappa_{WSM}$ predicted by application of mean (i.e., fixed) value of $\kappa_{inorg}$ and mean ± standard deviation of $\kappa_{WSOM}$ for 85 % using Eq. (2). The prediction captures the measured dependence of $\kappa_{WSM}$ on $f_{WSOM}$ at 85 % RH, supporting the importance of $f_{WSOM}$. As in the case of $\kappa_{WSM}$, dependence of $\kappa_{PM0.95}$ on the mass fraction of EOM in $PM_{0.95}$ ($f_{EOM}$) in the dehumidification branch was also observed for ≥60 % RH except

for 90 % RH (Figs. S12a and S12b). The result suggests that the mass fractions of organic components played an important role in the variation of the hygroscopicity of aerosol particles.

The relation between the degree of neutralization represented by $R_{A/S}$ and $\kappa_{WSM}$ at 20–90 % RH is also analyzed for the dehumidification branch (Figs. 8c and 8d). Although the correlations between $R_{A/S}$ and $\kappa_{WSM}$ were weak for ≥60 % RH, clearer

negative correlations were observed for <60 % RH. This result implies that the degree of neutralization is important to the variation of $\kappa_{WSM}$ under low RH conditions. The correlation was absent for $\kappa_{WSM}$ predicted from E-AIM versus measured $R_{A/S}$ ($r^2 \leq 0.33$ for RH ≤ 80 %). Therefore, the relation might be associated with the efflorescence behavior of inorganic components. Negative correlation at <60 % RH was observed ($r^2$: 0.58–0.77) even after excluding two samples with high relative abundances of sodium, which showed high ERH (Fig. 4c). Therefore, the deliquescence of ammoniated sulfate itself might be

related to $R_{A/S}$. An alternative explanation is that $R_{A/S}$ is related to water uptake by organics and/or their influence on the efflorescence of inorganic salts. Although the small amount of aerosol water at <60 % RH might not strongly affect the particle optical property, it might have an important role in chemical reactions in the particles. Therefore, the relation between inorganic composition and water uptake should be assessed further, in addition to the role of acidity itself in the reactions. As in the case of $\kappa_{WSM}$, the relation between $R_{A/S}$ and $\kappa_{PM0.95}$ in the dehumidification branch was analyzed (Figs. S12c and S12d). The result

suggests that the degree of neutralization of inorganic aerosol components is also important in low RH conditions.

In the humidification branch, moderately to highly negative correlations were found between $\kappa_{WSM}$ and $f_{WSOM}$ (and $\kappa_{PM0.95}$ and $f_{EOM}$) at ≥ 70 % RH (Figs. S13 and S14), indicating the contribution of WSOM to the water uptake of WSM (or $PM_{0.95}$), being similar to the case of the dehumidification branch. Moderate positive (or negative) correlations of $\kappa_{WSM}$ or $\kappa_{PM0.95}$ with $R_{A/S}$

were observed at 60 and 65 % RH (or 30 % RH), but for other RH conditions, correlation was not evident (Figs. S13 and S14). This result contrasts with the prediction of $\kappa_{WSM}$ from the E-AIM model (for inorganic components), which instead show moderate to high negative correlations with $R_{A/S}$ ratio, in particular at 50–70 % RH ($r^2 \geq 0.80$; Fig. S15). The strong positive correlations between E-AIM predicted DRH and $R_{A/S}$ ratio were also found (Fig. S16). They are expected to be responsible for negative correlation between model-based $\kappa_{WSM}$ and $R_{A/S}$. The reason for the contrasting results between the measurements

and the model prediction remains unclear. Further investigations of causal relations between neutralization and $\kappa_{WSM}$ (or $\kappa_{PM0.95}$) in the humidification mode are required. While the measurement uncertainty for DRH as seen for AS might be responsible for the discrepancy, the possible role of WSOM on DRH of inorganic salts should also be explored in future studies.





**4 Summary and Conclusions**

The composition of aerosols and the RH-dependent hygroscopic growth of aerosol components under the influence of the outflow from the Asian continent, as well as the air masses over the Pacific, were characterized based on analyses of submicrometer aerosol samples collected on filters in autumn 2015 in Okinawa, Japan. This offline analysis compensated for

online analysis in terms of the quantification and characterization of water-soluble components and $PM_{0.95}$ (50 % cutoff diameter: 0.95 μm), and of the measurement of the hygroscopic growth as a function of relative humidity. This study characterized the RH-dependent hygroscopicity of submicrometer aerosols and their chemical components, in particular organics, in the outflow region of East Asia. Another important point is that results from offline analyses were compared to those collected using online methods to assess the consistency of the results from the two different approaches.

The analysis of $PM_{0.95}$ samples collected on filters showed the dominance of sulfate, which is quantitatively consistent with the chemical composition analysis based on online analysis using AMS and COSMOS. Offline analysis showed high proportions of WSOM (93 %) in EOM and high O:C of WSOM and EOM ($0.84\pm0.08$ and $0.79\pm0.08$, respectively), all of which indicate the aged nature of the studied aerosol and characterize long-range transported aerosols off the coast of East

Asia. The temporal variation of the ammonium-to-sulfate molar ratio, assuming that sulfate was neutralized preferentially by sodium ions ($R_{A/S}$), was obtained from the offline analysis. The result demonstrates that air masses from the Asian continent transported more acidic and aged aerosols and that sodium played a role in neutralizing the sulfate from maritime air masses.

The RH dependence of hygroscopic growth in humidification and dehumidification branches was inferred for water-soluble

components from filter samples. At 40, 60, and 85 % RH, the $\kappa$ values for WSM ($\kappa_{WSM}$) in the humidification (dehumidification) branch were, respectively, $0.21\pm0.10$ ($0.45\pm0.17$), $0.30\pm0.13$ ($0.57\pm0.04$) and $0.51\pm0.04$ ($0.50\pm0.03$). The $\kappa$ values for $PM_{0.95}$ ($\kappa_{PM0.95}$) were also calculated. At 75–85 % RH, the $\kappa$ values of WSOM and EOM in the dehumidification branch were estimated respectively as 0.19–0.22 and 0.17–0.20. The WSOM was estimated to have contributed to 10–12 % of the water uptake of WSM and $PM_{0.95}$. The dependences of the $\kappa_{WSM}$ (or $\kappa_{PM0.95}$) on the mass fractions of WSOM in WSM

(or $PM_{0.95}$), and the $R_{A/S}$ (at ≤50 % RH) in the dehumidification mode were inferred from their negative correlations. In the humidification mode, whereas the mass fractions of WSOM was suggested to be important for both $\kappa_{WSM}$ and $\kappa_{PM0.95}$ at RH ≥ 70 %, the relation between $\kappa_{WSM}$ (or $\kappa_{PM0.95}$) and $R_{A/S}$ in the low RH range was not evident despite of the apparent relation in the case of $\kappa_{WSM}$ predicted from E-AIM. The dependence of $\kappa_{WSM}$ and $\kappa_{PM0.95}$ on the fractions of WSOM and $R_{A/S}$ suggests the importance to understand the temporal variability of the aerosol hygroscopicity at the receptor region of East Asian outflow,

which includes a long-term trend under the condition of the large decrease of Chinese $SO_2$ emissions in recent years (Zheng et al., 2018).





The hygroscopicity parameter values of PM$_{0.95}$ at 85 % RH from offline methods were close to earlier reported values from online hygroscopicity measurements performed during the field campaign. Results obtained from this study extended the characterization of the studied aerosols by online analysis (≤200 nm), toward the mass/volume based mean diameter of the submicrometer aerosols. On the other hand, the similarity of the hygroscopicity parameter values from offline and online

methods suggest the propriety of the offline method on aerosol hygroscopicity analysis, at least for remote sites at which the aerosols are aged and semi-volatile ammonium nitrate is not abundant. This finding encourages further studies of the hygroscopicity of aerosol components, particularly OA, the hygroscopicity of which is not yet characterized well. Given that precise analysis of the hygroscopicity of OA is not easy based on online analyses, the offline approach is useful for better understanding of the relation between chemical structure, sources and hygroscopicity of WSOM and other organic components,

because of the richness of information from the AMS spectra. For example, the hygroscopicity of humic-like substances and other organic fractions and their contributions to total particulate matter are worth elucidating by the extension of the approach of this study.

**Abbreviations and symbols**

$a_w$      water activity

$\varepsilon_{EC/PM0.95}$      volume fraction of EC in PM$_{0.95}$

$\varepsilon_{EOM/PM0.95}$      volume fraction of EOM in PM$_{0.95}$

$\varepsilon_{WISOM/EOM}$      volume fraction of WISOM in EOM

$\varepsilon_{WSIM/PM0.95}$      volume fraction of WSIM in PM$_{0.95}$

$\varepsilon_{WSIM/WSM}$      volume fraction of WSIM in WSM

$\varepsilon_{WSOM/EOM}$      volume fraction of WSOM in EOM

$\varepsilon_{WSOM/WSM}$      volume fraction of WSOM in WSM

$\rho_w$      density of pure water

$\sigma$      surface tension at the solution–air interface of a liquid particle

$\kappa$      hygroscopicity parameter

$\kappa_{EC}$      $\kappa$ of EC, which is equal to zero

$\kappa_{EOM}$      $\kappa$ of EOM

$\kappa_{inorg}$      $\kappa$ of WSIM

$\kappa_{online}$      $\kappa$ of ambient aerosol particles at 85 % RH obtained through on-site measurement

$\kappa_{PM0.95}$      $\kappa$ of PM$_{0.95}$

$\kappa_{WSM}$      $\kappa$ of WSM

$\kappa_{WSOM}$      $\kappa$ of WSOM





| | | |
|---|---|---|
| AMS | (high-resolution time-of-flight) aerosol mass spectrometer | |
| AS | ammonium sulfate | |
| BC | black carbon | |
| COSMOS | continuous soot monitoring system | |
| 5 | CPC | condensation particle counter |
| | $d_{dry}$ | dry particle diameter, which is 100 nm for this study |
| | DMA | differential mobility analyzer |
| | DRH | deliquescence RH |
| | $d_{wet}$ | wet particle diameter: the product of $g_f$ and $d_{dry}$ |
| 10 | E-AIM | online Extended AIM Aerosol Thermodynamics model |
| | EC | elemental carbon |
| | EOC | extracted organic carbon |
| | EOM | extracted organic matter |
| | ERH | efflorescence RH |
| 15 | $g_f$ | hygroscopic growth factor |
| | H:C | atomic ratio of H to C |
| | H/V-TDMA | hygroscopicity and volatility tandem differential mobility analyzer |
| | HTDMA | hygroscopicity tandem differential mobility analyzer |
| | IC | ion chromatograph |
| 20 | $M_w$ | molar mass of pure water |
| | O:C | atomic ratio of O to C |
| | OA | organic aerosol |
| | OC | organic carbon |
| | OM:OC | mass ratio of organic matter to organic carbon in the organic aerosol component |
| 25 | $PM_{0.95}$ | subset of aerosol particles with diameters <0.95 μm |
| | $PM_1$ | subset of aerosol particles with diameters <1 μm |
| | $R$ | universal gas constant |
| | $r^2$ | coefficient of determination between two variables |
| | $R_{A/S}$ | molar ratio of ammonium to the remaining sulfate after preferentially being neutralized by sodium |
| 30 | $R_{A/S}{'}$ | molar ratio of ammonium to sulfate |
| | RH | relative humidity |
| | SMPS | scanning mobility particles sizer |
| | $T$ | absolute temperature |



TOC      total organic carbon

WISOC  water-insoluble organic carbon

WISOM water-insoluble organic matter

WSM     water-soluble matter

WSIM   water-soluble inorganic matter

WSOC   water-soluble organic carbon

WSOM  water-soluble organic matter

**Data availability.** All of the finally derived data supporting the findings of this study are available in the article or in its
supporting information file.

**Author contributions.** MM, HF, and YD designed the experiments with contributions from KS and SH. HF, YD, YH, and
MM performed them with contributions from KS, YM, ET, DD, TN, ST, HX, FL, and SO. YD, MM, and HF analyzed the
data with contributions from KS, YM, ET, DD, KK, TN, ST, MC, HT, SO, YK, AT, and SH. YD and MM prepared the
manuscript with contributions from HF, YM, DD, SO, and AT.

**Competing interests.** Author Kimitaka Kawamura is a member of the editorial board of the journal.

**Acknowledgments.** We thank the staff of the Cape Hedo Atmosphere and Aerosol Monitoring Station, National Institute for
Environmental Studies, Japan, for the use of the study site. We also thank Atsushi Matsuki for the use of the water-based CPC,
Petr Vodička for expert discussions on the OC/EC data analysis and comments on the paper, Martin Irwin for the field BC
observation, and Naga Oshima for the quality control of online BC data. We also thank Zhou Ruichen and Shuhei Ogawa for
the calculation of the resident time of sample aerosols after humidification in the HTDMA. We acknowledge the NOAA Air
Resources Laboratory (ARL) for providing the HYSPLIT transport and dispersion model. This study was supported in part by
JSPS KAKENHI Grant Numbers JP19H04253 and JP18K19852 and the Environment Research and Technology Development
Fund (JPMEERF20202003) of the Environmental Restoration and Conservation Agency of Japan, and was in part performed
under the joint research program of Institute for Space–Earth Environmental Research, Nagoya University. YD thanks the
current affiliation for supporting her to process data and write this paper.

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









**Figure 1: (a) Mass concentrations of chemical components in PM$_{0.95}$ and (b) their mass fractions from offline analyses. (c) Mass concentrations of PM$_1$ components and (d) number–size distributions of aerosol particles from online analyses. A related online BC data has been published by Koike and Oshima (2018). The pie charts in panels (a) and (c) show the mean fractions of each compound calculated from their mean mass concentrations based on offline and online analyses, respectively. The pattern wedge in the pie chart in panel (a) represent the total mass fraction of K$^+$, Mg$^{2+}$, Ca$^{2+}$, Cl$^-$, and NO$_3^-$.**

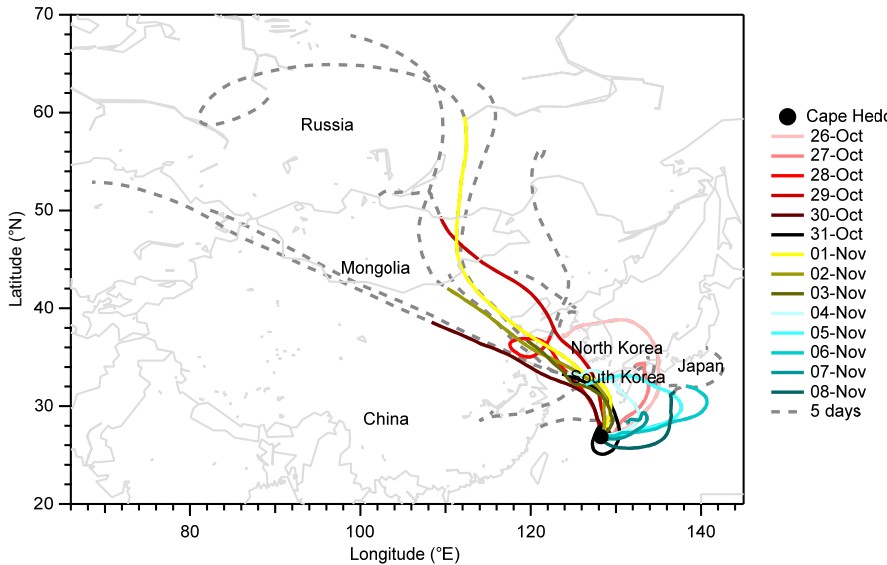

**Figure 2: Daily air mass trajectories arrived at 500 m above the observation site at 1400 JST. The solid-colored lines represent the three-day trajectories. The solid-colored lines together with their respective extended gray dash sections indicate the five-day trajectories. The solid circle represents the observation site location. The map is based on GSHHG 2.3.4; the shoreline polygon data in crude resolution is used. Trajectories were produced using NOAA HYSPLIT atmospheric transport and dispersion modeling system (Draxler and Hess, 1998).**



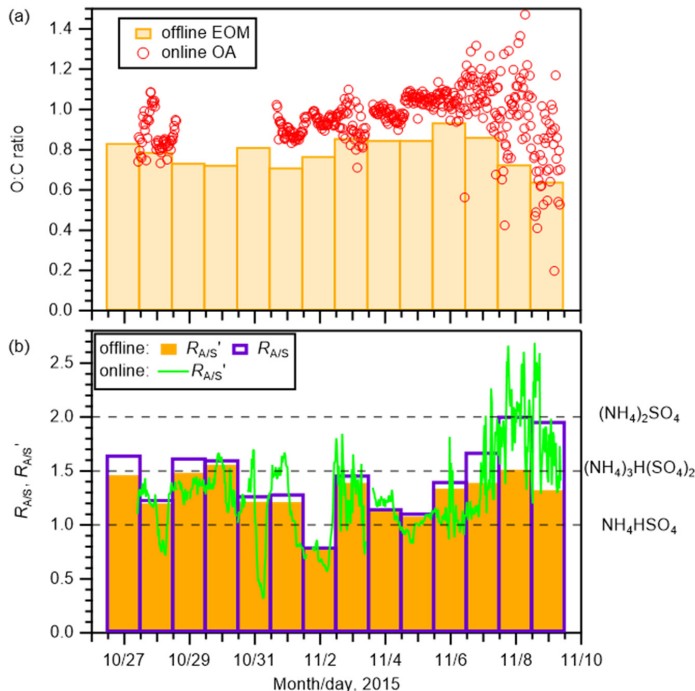

**Figure 3: (a) The O:C of EOM from the offline analysis of PM$_{0.95}$ samples (bars) and that of OA from the online AMS analysis (open circles). (b) The degree of neutralization of the remaining sulfate (after preferentially being neutralized by sodium; Text S7) by ammonium from the offline analysis of PM$_{0.95}$ samples ($R_{A/S}$; box bars); the degree of neutralization of sulfate by ammonium (without**
5 **considering sodium) ($R_{A/S}'$) from the offline analysis (solid bars) and the online AMS analysis (solid line). The $R_{A/S}'$ values of NH$_4$HSO$_4$, (NH$_4$)$_3$H(SO$_4$)$_2$, and (NH$_4$)$_2$SO$_4$ are shown as dashed lines in panel (b). In panel (a), the missing of O:C during 28–31 October is due to the malfunction of chopper in the W-mode online measurement.**





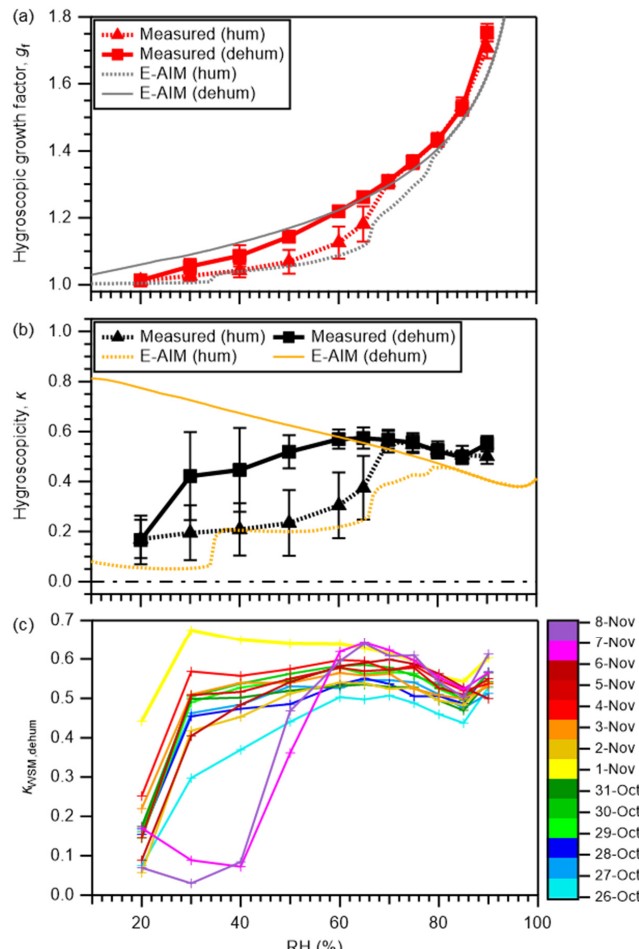

**Figure 4:** The mean of (a) measured and E-AIM predicted $g_f$ for WSM particles as a function of RH and (b) measured and E-AIM predicted $\kappa_{WSM}$ as a function of RH. (c) The $\kappa$ values for WSM from respective PM$_{0.95}$ samples in dehumidification (dehum) branches. In panels (a) and (b), results from both humidification (hum) and dehumidification branches are presented. In the predictions in panels (a) and (b), water retained by WSOM are not considered. Results obtained for the dehumidification branch were obtained by assuming that no solid is formed under any RH condition.



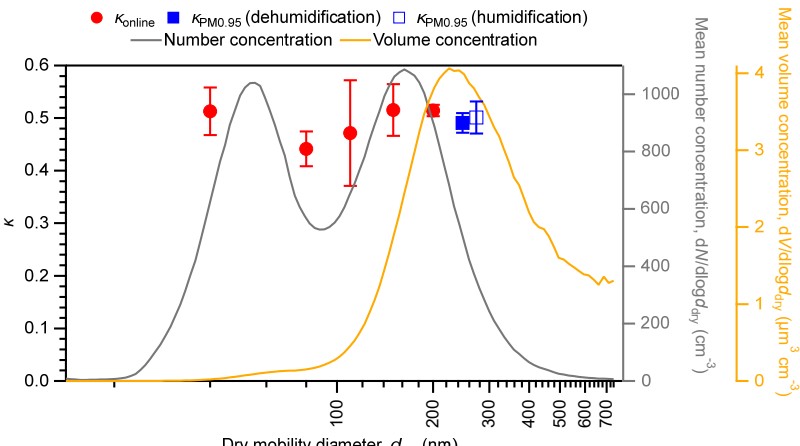

**Figure 5: Comparison of $\kappa$ for PM$_{0.95}$ ($\kappa_{PM0.95}$) in humidification and dehumidification branches, and $\kappa$ from online analyses ($\kappa_{online}$) during the same observation period. Markers and whiskers respectively represent mean values and standard deviations. The mean number/volume concentrations of atmospheric aerosols are also shown.**

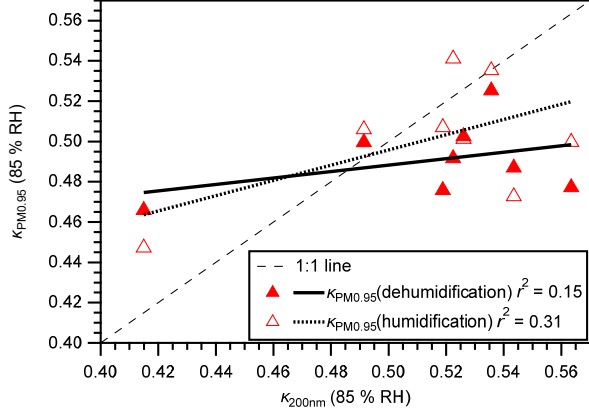

**Figure 6: Comparison of calculated $\kappa$ for PM$_{0.95}$ ($\kappa_{PM0.95}$) in humidification and dehumidification branches and $\kappa$ for 200 nm atmospheric particles from online HTDMA analysis. Regression lines and 1:1 line are also presented. It is noteworthy that online**





data only represent less than 2 % of the aerosols in a day, whereas offline data represent more than 95 % of aerosols in a day (10 am – 10 am).

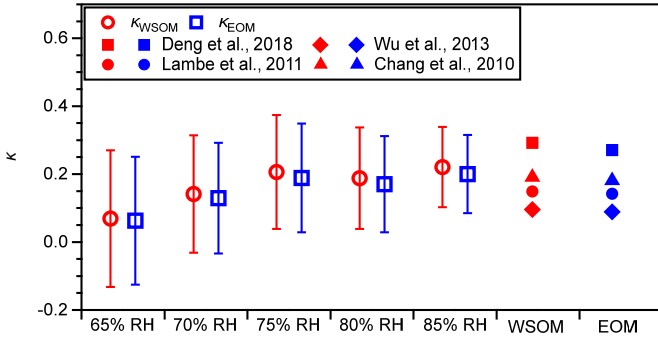

**Figure 7: Estimated hygroscopicity parameter values of WSOM ($\kappa_{WSOM}$) and EOM ($\kappa_{EOM}$) under different RH conditions in dehumidification branches. Open markers present mean values. Whiskers present standard deviations of the related mean value. The hygroscopicity parameter values of WSOM and EOM predicted based on regression lines reported for the hygroscopicity parameter values of organics and its O:C (Chang et al., 2010; Lambe et al., 2011; Wu et al., 2013; Deng et al., 2018), where mean values of the O:C of WSOM (0.84, red markers) and of EOM (0.79, blue markers) were used.**

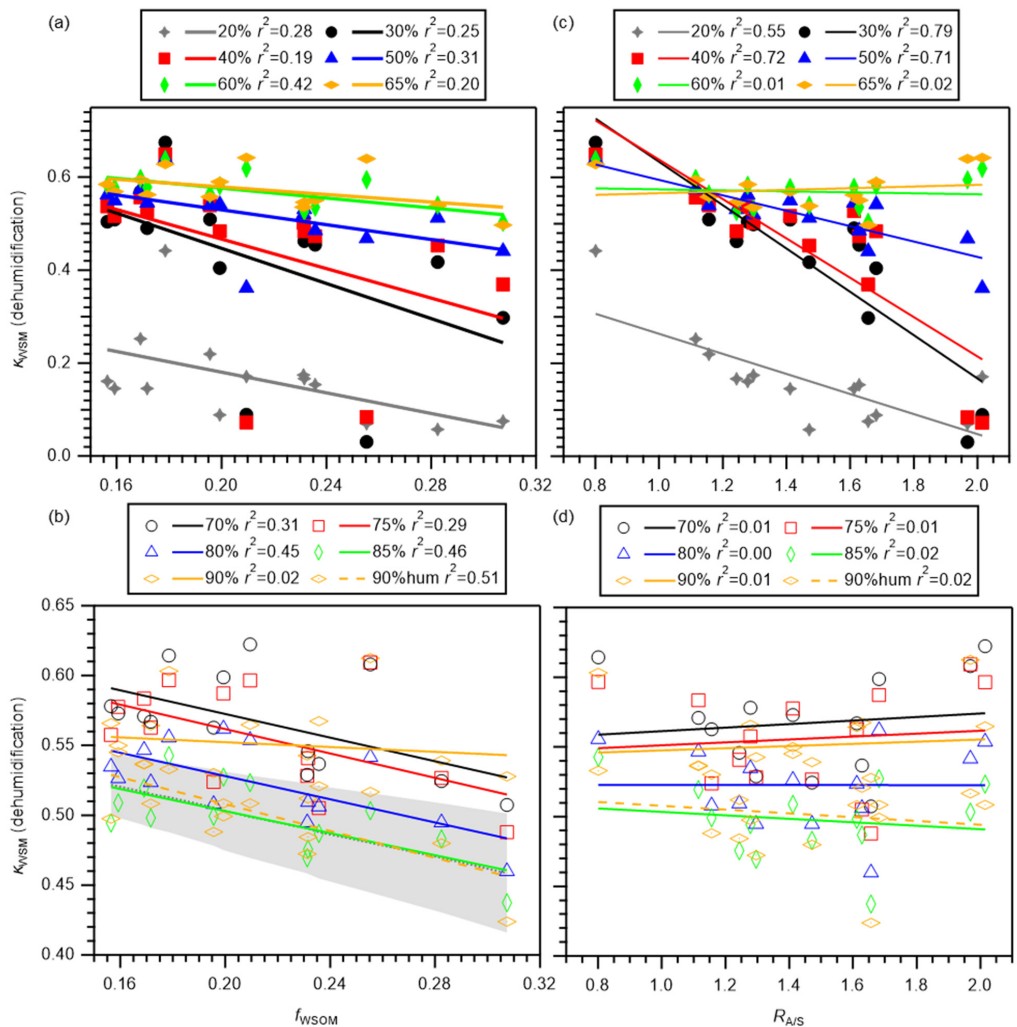

**Figure 8: The $\kappa_{WSM}$ values in the dehumidification branch versus (a and b) the mass fractions of WSOM in WSM ($f_{WSOM}$) and (c and d) the ammonium-to-remaining sulfate molar ratio ($R_{A/S}$) from the offline analysis. Panels a and c present results obtained at 20, 30, 40, 50, 60, and 65 % RH. Panels c and d present results obtained at 70, 75, 80, 85, and 90 % RH. The $\kappa_{WSM}$ at 90 % RH in the humidification branch (90%hum) was also compared to (b) $f_{WSOM}$ and (d) $R_{A/S}$. The shaded area in panel (b) show $\kappa_{WSM}$ predicted by application of mean value of $\kappa_{inorg}$ (0.59) and mean ± standard deviation of $\kappa_{WSOM}$ for 85 % RH (0.22±0.12) using Eq. (2). Coefficients of determination $r^2$ are also presented.**