# Peer review of "Offline analysis of the chemical composition and hygroscopicity of sub-micrometer aerosol at an Asian outflow receptor site and comparison with online measurements"

_Atmospheric Chemistry and Physics, 2021_

## Author Response (AR1)

**(The texts in red were added after we submitted the final response of the open discussion.)**

We appreciate valuable comments on our manuscript by the reviewers. Our responses are listed below.

**Response to Anonymous Referee #1**

This manuscript presents aerosol chemical composition and hygroscopicity from offline measurements and comparison with online analysis in Okinawa based on a combination of instruments. The RH-dependent hygroscopicity of sub-micrometer aerosols and their chemical components are investigated. The measurements and data have been made carefully and then the interesting results are presented, especially the comparison between the two methods is helpful to understand whether the offline analysis can be used as an alternative method for aerosol hygroscopicity studies.

However, I have some major concerns about the bias of the offline analysis. And it could be more cautious and thorough when interpreting some results. In addition, there are also some editing issues need that have to be addressed. The authors thus need to make a careful revision and correction to improve the overall quality of the paper for publication in the journal. I would recommend the editor to reconsider the papers only after a major revision by the authors.

**Major Comments:**

1. Page 4, Line 30, the samples were stored in freezers after sample collection. How to ensure that compounds do not undergo some physical and chemical changes such as condensation and degradation during storage? Have you considered the offline analysis bias caused by storage?

As reported in this paper we found good agreements between online and offline analyses, suggesting that the degradation of samples during storage is not critical in view of the objective of this study. Because the degradation of some highly reactive compounds such as persistent radicals (Alpert et al., 2021) is not ruled out, it is now addressed as a point to consider when a filter-based approach is used for other types of aerosol studies. Although to

check possible changes in the composition and property of the samples is an option, the samples we used are not new and further assessment of this point is beyond this study. The original expression:

"After sample collection, the filter samples were stored in freezers before analysis."

**has been modified to**

"After sample collection, the filter samples were stored in freezers for around nine months or longer before the analyses. Possible degradation of some highly reactive compounds such as persistent radicals (Alpert et al., 2021) is not assessed in this study." (Page 4 lines 30–32)

**2. Page 9, the authors concluded good agreement between offline and online analyses from the high positive correlations of mass concentrations, but the $R^2$ of the mass fractions of sulfate, organics, ammonium, and EC (BC) from offline and online analysis were low. The authors should try to explain this.**

The fact that the  $r^2$  for the mass fractions of offline and online analyses were lower than those of the mass concentrations is in line with the propagation of uncertainty. If the mass concentrations of sulfate, organics, ammonium, and EC (or BC) are assumed to be uncorrelated, the relative uncertainty of their mass fractions can be represented as:

$$\frac{\delta f_i}{f_i} = \sqrt[2]{\left(\frac{\delta a_i}{a_i}\right)^2 + \frac{(\delta a_1)^2 + (\delta a_2)^2 + (\delta a_3)^2 + (\delta a_4)^2}{(a_1 + a_2 + a_3 + a_4)^2}} \quad (i = 1, 2, 3, 4)$$
(AR1)

where,  $a_1$  (or  $f_1$ ),  $a_2$  ( $f_2$ ),  $a_3$  ( $f_3$ ), and  $a_4$  ( $f_4$ ) are the mass concentrations (mass fractions) of sulfate, organics, ammonium, and EC (or BC), respectively, and the expression  $\delta X$  denotes the uncertainty of X. It can be seen that the relative uncertainties of the mass fractions represented by equation AR1 are greater than those of the mass concentrations for same compounds. Besides, the magnitudes of the ranges of the mass fractions relative to the averages are smaller than those of the mass concentrations (Fig. S7 in the revised SI), suggesting the stronger contributions of the uncertainty on  $r^2$ .

We added the following explanation to the main text:

"Note that the  $r^2$  of the mass fractions between offline and online analyses were lower than those of the mass concentrations. It may be in part because the mass fractions of respective components are influenced by the uncertainties of the mass concentrations of respective components and also by those of the summed concentrations. In addition, smaller variations of the mass fractions as compared to the mass concentrations may result in larger contributions of the uncertainties to  $r^2$ ." (Page 10 lines 8–12)

**In addition, the average mass concentrations of sulfate, organics, and ammonium from online measurements were lower, but BC from online analysis was almost equal to those from offline analysis. What is the reason for this?**

Possible reasons for the lower average mass concentrations of sulfate, organics, and ammonium from online AMS measurements were already explained in Sect. 3.1. Because the online measurement of BC was not made by the AMS, a different degree of agreement with the offline analysis (of EC in this case) for the average mass concentration is not contradict with our explanation.

3. Section 3.2, the authors should add more measured results from offline analysis and compare these with previous studies since many previous online analyses have tried to derive the hygroscopicity in a larger accumulation size above 300 nm. It is best to add more discussions in these paragraphs.

The last paragraph in Sect. 3.2 has been modified to two paragraphs. A brief explanation on the alternate methods of the hygroscopicity measurement for >500 nm particles (above the upper measurement limit of basic HTDMA; Tang et al., 2019), and a comparison with previous results about the hygroscopicity of >300 nm particles is now explained in the new final paragraph of the section:

"Most HTDMA are not applicable to the measurements of the hygroscopicity of dry particles larger than 500 nm (Tang et al., 2019), although techniques to measure >500 nm particles have also been developed, for example based on the usage of optical particle counters (Sorooshian et al., 2008; Tang et al., 2019). Comparison with the  $\kappa$  of ambient particles with  $d_{dry}$  of 300 nm or larger in previous studies shows that the mean of  $\kappa$  at 85 % RH for PM0.95 from our study (0.47) is larger than the mean  $\kappa$  values of 300–360 nm particles in an urban site (0.32–0.33; Kawana et al., 2016) and a forest site (0.34–0.40; Kawana et al., 2017; Deng et al., 2019) in East Asia." (Page 14 lines 16–21)

4. Page 13, Line 19, "Results obtained from ... indicate that offline aerosol hygroscopicity analysis can be used as an alternative method", but the coefficients of determination between offline and online results were low (Fig. 6), even less than 0.5. The authors should try to address this and make it convincing.

Considering the facts that the particle hygroscopicity stayed high and the variation was small during the field campaign, we regard that the finding of the positive correlation between the online and offline data is remarkable. In addition, all data were within ~15% from the 1:1 line (the updated version of Fig. 6). In addition to measurement uncertainties, possible factors that lower the coefficients of determination are: (1) the offline/online data were for different size ranges, and (2) online data represent less than 2 % of the aerosols in a day whereas offline data represent more than 95 % of aerosols in a day. An explanation about these points has been added to the main text as follows.

The original expression: "The  $\kappa_{PM0.95}$  and  $\kappa_{online}$  at 85 % RH for 200 nm particles are compared for the days when both data are available (Fig. 6), which showed moderate positive correlations:  $r^2$  of 0.15 and 0.31 respectively for dehumidification and humidification branches."

**has been modified to**

"Furthermore, the  $\kappa_{PM0.95}$  and  $\kappa_{online}$  at 85 % RH for 200 nm particles are compared for the days when both data are available (Fig. 6). Moderate positive correlations were found ( $r^2$  of 0.37 and 0.42 respectively for dehumidification and humidification branches), although the particle hygroscopicity stayed high and the variation was small during the campaign. In addition, all data are within ~15 % from the 1:1 line (Fig. 6). The absence of strong correlations may be because the offline/online data were for different size ranges, and because online data only represent less than 2 % of the aerosols in a day whereas offline data represent more than 95 % of aerosols in a day." (Page 13 line 30–page 14 line 2)

5. Fig. 7, comparing the hygroscopicity parameter of WSOM and EOM between this work and previous experiments needs to be shown to be much more rigorous. Unless the authors can explicitly show that the experiment setup and estimation method were identical between the different experiments, they cannot make a like-for-like comparison between the different experiments. In addition, the RH conditions of these previous studies should be included in the figure.

The RH conditions of the previous study have been added both in the main text and in the caption of Fig. 7. We regard our comparison is meaningful because the difference in  $\kappa_{\text{org}}$  between supersaturated and subsaturated conditions may be small (Liu et al., 2018; Kuang et al., 2020). The explanation about the comparison has been revised as follows:

"The mean O:C of WSOM and EOM were 0.84 and 0.78, from which  $\kappa_{WSOM}$  and  $\kappa_{EOM}$  were estimated respectively to be 0.19 and 0.18 based on the reported regression lines between  $\kappa_{org}$  and O:C under sub-saturated and supersaturated conditions (0.42 ± 0.04 % supersaturation (Chang et al., 2010); 0.1–1.5 % supersaturation (Lambe et al., 2011); 90 % RH (Wu et al., 2013); 0.11–0.80 % supersaturation (Deng et al., 2018)) (Fig. 7). Although there may be some difference in the  $\kappa_{org}$  between sub-saturated and supersaturated conditions, we regard our comparison is meaningful because the difference could be small (Liu et al., 2018; Kuang et al., 2020)." (Page 15 lines 13–19)

**Minor Comments:**

**1. Page 5, Line 30, The residence time in the heated region should be compared with that for other systems.**

Samples were not heated. According to Duplissy et al. (2009), the residence time of the particles in the equilibrium RH region should be 10-40 s. Thus, the original expression "The residence time from the outlet of the NH2 to the inlet of DMA2 was calculated as 13 s." has been modified to "The residence time from the outlet of the NH2 to the inlet of DMA2 was calculated as 13 s, which is close to the lower end of the recommended range of 10-40 s by Duplissy et al. (2009)." (Page 6 line 2)

**2. A schematic of the experimental set-up would help.**

Schematics for key instruments have been added as follows.

Figure S2: Schematics of the experimental set-up for (a) the offline hygroscopic growth measurement and (b) online AMS/SMPS measurements. Flow rates are in the unit of L min-1. DMA: differential mobility analyzer; CPC: condensation particle counter; NH: Nafion humidifier; AMS: aerosol mass spectrometer.

Figure S2 is referred in the main text. (Page 5 line 18 and page 7 line 31)

**3. The retrieval method of the HTDMA data should be included in section 2.2.**

We regard that it is not necessary to address it in detail because the retrieval method has been described by Mochida et al. (2010). We have added more explanation to the original description: the original expression "..., which were retrieved using the Twomey algorithm as described by Mochida et al. (2010)." has been modified to "..., which were retrieved using

the Twomey algorithm with consideration of transfer functions of the two DMAs as the work in Mochida et al. (2010) ("T" in Eq. S2 in the reference should be omitted) but with modified data bins." (Page 6 lines 5–6)

**4. Fig. 1, add the mass fractions from online analysis.**

A figure showing the mass fractions from online analysis has been added to Fig. 1. The caption of Fig. 1 has also been modified accordingly.

**5. Fig. S6, looks like one point is missing in the Fig. S6(c).**

The data from 7-Nov and 8-Nov are very similar. Using a different marker in Fig. S6, which is Fig. S7 in the revised SI, solved the problem.

**6. I suggest adding more 'raw' measurement time series (e.g. in the supplementary information). For example, time series of hygroscopicity parameter over the experiment.**

For online measurements, the time series of the chemical composition and number-size distribution of aerosols can be found in Fig. 1. The time series of the mean  $g_f$  of aerosol particles with different dry diameters have been added as Fig. S14, which is referred in the main text as "The time series of the mean  $g_f$  of particles with different dry diameters are presented in Fig. S14.". (Page 13 lines 28–29)

**7. Page 10, I was lost from line 7 to 11 when the aerosol concentrations and fractions are discussed. Is the discussion still based on Fig. 2 or other figures in the manuscript?**

The discussion is based on Figs. 1a, 1b, and 2, and Table S3. The original expression:

"A comparison between air mass trajectories and aerosol concentration data shows that maritime air masses during 6–8 November are characterized by lower aerosol mass concentrations, but higher mass fractions of sodium ( $\geq$ 5 %) than the other days influenced by continental air masses. The mean mass concentrations of sulfate, WSOM, ammonium, and EC from offline analyses during 6–8 November were, on average, 1/6, 1/5, 1/5, and 1/2 of those during other days, respectively, whereas the mean mass concentration of sodium during the period (0.07 µg m-3) was similar to that of other days (0.06 µg m-3)."

has been modified to:

"A comparison between air mass trajectories and aerosol concentration data shows that maritime air masses during 6–8 November are characterized by lower aerosol mass concentrations, but higher mass fractions of sodium ( $\geq$ 5 %) than the other days influenced by continental air masses (Figs 1a, 1b and 2). The mean mass concentrations of sulfate, WSOM, ammonium, and EC from offline analyses during 6–8 November were, on average, 1/6, 1/5, 1/5, and 1/2 of those during other days, respectively, whereas the mean mass concentration of sodium during the period (0.07 µg m-3) was similar to that of other days (0.06 µg m-3) (Table S3).". (Page 10 lines 23–28)

**Response to Anonymous Referee #3**

This manuscript performed offline analysis of both chemical composition and hygroscopicity of submicrometer particles and then compared the results with those from online measurements. In traditional view, the offline filter analysis is tedious and time-consumed but this work presents a very interesting approach to compliment the online results. It is worthy of exploration since it might provide a more affordable way to measure the aerosol hygroscopicity, especially when online instruments are not available and offline analytical analysis can be easily accessible. Although it is found very interesting, the manuscript needs to be improved and the following issues should be fully resolved, before it can be publishable.

**Major comments:**

1. Although the authors made great efforts to elaborate how to avoid artifacts and those artifacts indeed can be minimized. However, the authors should also give some more details on how errors are propagated, for example, the errors generated from exaction of organics from filters and the nebulization of organics, subsequent growth measurement etc. To what extent, the errors are associated with the offline analysis presented in this study?

The errors associated with the offline hygroscopicity analysis (of organics) have been assessed from the following four aspects:

Filter sampling artifact: This point was already addressed in the second paragraph of Sect.
 Based on the comparison of the mass concentrations of sulfate, organics, and ammonium between offline and online analyses.

2) The extraction efficiency of organics: As presented in Figure S6, fair agreement between the mass concentrations of extracted OC from the offline AMS analysis and those of OC from the thermal analysis supports high extraction efficiency of organics by the solvent extraction. The good performance of the quantification by the offline AMS with the use of phthalic acid as internal standard is presented in Chen et al. (2016). An explanation to this point has been added to Sect. 2.1:

"The fair agreement between the mass concentrations of extracted OC from the offline AMS analysis, the performance of which has been well validated in the work of Chen et al. (2016), and those of OC from the thermal analysis support the high extraction efficiency of organics by the solvent extraction." (Page 5 lines 12–15)

3) Nebulization artifact: If dried particles entering the first DMA in the HTDMA are nonspherical or porous, it introduces bias in the calculation of hygroscopicity parameter  $\kappa$ . While  $g_f$  smaller than unity was reported and interpreted as a result of restructuring of non-spherical particles (Gysel et al., 2004, Jung et al., 2011), the  $g_f$  of nearly unity or greater for WSM particles in our study (>0.997) means that such phenomenon was not evident. It may owe to the humidification using NH1 followed by drying in diffusion driers prior to the HTDMA analysis, which may avoid fast water evaporation and formation of cracks or cavities (Gysel et al., 2004). This explanation has been added to the main text and Text S3.

In the main text, "While  $g_f$  smaller than unity was reported and interpreted as a result of restructuring of non-spherical particles (Gysel et al., 2004, Jung et al., 2011), the  $g_f$  of nearly unity or greater for WSM particles in our study (>0.997) means that such phenomenon was not evident. It may owe to the humidification using NH1 followed by drying in diffusion driers prior to the HTDMA analysis, which may avoid fast water evaporation and formation of cracks or cavities (Gysel et al., 2004)." has been added to Sect. 3.2. (Page 11 lines 25–29)

In Text S3, the original expression "The considerable disagreement at low RH might be attributable to the assumption of perfectly spherical and non-porous solid AS particles, and might also be attributable to bias of the E-AIM model and uncertainty of the RH measurements." has been deleted. Instead, the following explanation has been added.

"Note that the determined values of  $g_f$  were nearly unity or greater at all RH for both AS (>0.996) and WSM (>0.997) (Fig. S3 and Sect. 3.2). Therefore, there is no indication of non-sphericity of dried particles in the HTDMA (Gysel et al., 2004; Jung et al., 2011)."

Furthermore, the plots for 20 and 30 % RH in the humidification branch and for 20 % RH in the dehumidification branch in Fig. S3(b) have been corrected.

4) Uncertainty of hygroscopic growth measurement: The uncertainty that may be strongly associated with the uncertainty of RH has been addressed based on the assessment using ammonium sulfate particles (Text S3). Another source of uncertainty during the growth measurement could be the evaporation of particles. However, the evaporation artifact caused by ammonium nitrate should not be important in this study because the concentrations in the samples were low. Furthermore, the  $g_f$  of nearly unity or greater for WSM particles in our study (>0.997) means that such phenomenon was not evident, as explained above.

2. Recent literatures present a growing interest in the effects of surfactants on modification of surface tension of the particles and hence affect the hygroscopicity of the aerosol particles, including measurement inland, at sea, etc. The use of surface tension of pure water neglecting the above-mentioned effect and apparently it is problematic. How organics affect the surface properties of the particles and what surface tension range of values would be estimated from this study? Could the authors elaborate a bit more on this aspect?

It is difficult to estimate the actual surface tension values of the droplets in the HTDMA. If the surface tension of WSM droplets in the HTDMA was assumed to be 30 % lower than that of pure water (similar to the largest decrease experimentally observed in the study by Facchini et al. (1999), the calculated that  $\kappa_{WSM}$  is 0.8–3.4 % lower than that with the assumption of the surface tension of pure water. A brief explanation on this point has been added to the text:

"Note that the assumption of 30 % lower  $\sigma$  than that of water (Facchini et al., 1999) results in a slight decrease in the calculated  $\kappa_{WSM}$  (0.8–3.4 %), which provides guide in the uncertainty associated with surface tension." (Page 6 lines 16–18)

3. It is quite interesting to see how the AMS spectra from extracted WSM are different from the average online AMS spectra for the same time period. Could the authors do some comparisons and present some results for the spectra between the two methods? What would make those differences?

The comparisons between offline and online analyses based on mass spectra of organics have been added as Text S8, which is referred in the main text as follows:

"Further comparisons between offline and online analyses based on mass spectra of organics are presented in Text S8." (Page 11 lines 10–11)

4. Following the above question, how if the authors performed some factorization analyses based on the offline AMS method and then compared with those from online AMS data if available. Do they have the similar results?

Because the source apportionment of organic aerosol is beyond the scope of this study, we are not applying the PMF analysis for the AMS data. Given that the organic aerosol should have been aged during the long-range transport, it is questionable if the PMF analysis works very well. A previous study using a PMF analysis for organic aerosol in spring season from the same site showed that low-volatile oxygenated organic aerosol accounted for 76 % of total organic aerosol (Yoshino et al., 2021).

**Minor:**

1. Quite a few instruments were employed to perform the offline analyses, corresponding to many chemical components as presented in the paper. It might be beneficial to the readers if the authors can provide a table showing all the measured components with their measurement techniques.

A summary of all offline measurements has been added as Table S4, which is referenced in the main text as "Table S4 summarizes all offline measurements." (Page 7, line 23).

**2. Line 19 on p2, "serve as"?**

The phrase "serve" has been changed to "serve as". (Page 2 line 20)

**3. Line 11 on p7, "were obtained"?**

The phrase "were found" has been changed to "were obtained". (Page 7 line 19)

**4. Line 15 on p9, explain why values of 1.8 and 1.2 were used.**

The conversion factors of 1.8 and 1.2 were chosen according to Müller et al. (2017a) and references therein. This information has been added to the text. (Page 9 line 24)

**5. Line 29 on p11, "under highly acidic conditions..."?**

According to the comment, the original sentence "...can be enhanced by highly acidic conditions (Sect. 3.1) and/or the presence of WSOM..." has been modified to "...can be enhanced under highly acidic conditions (Sect. 3.1) and/or in the presence of WSOM..." "Gysel et al., 2014" in the original sentence was a typo, and has been corrected to "Gysel et al., 2004". (Page 12 lines 17–18)

**6. Line 15 on p13, how large is large? Do you have a criterion?**

Large particles referred to particles that are in the accumulation mode. Therefore, the original expression "the hygroscopicity of large aerosol particles might be more important." has been modified to "the hygroscopicity of large aerosol particles in the accumulation mode might be more important." (Page 14 line 14)

7. Line 1 on p14, "at a supersite"; Line 4, "suggest the importance on considering" or something better instead of using "to"; Line 14, "estimated to be"? Lines 31-32, this sentence seems awkward, please change it.

The phrase "in a supersite" has been changed to "at a supersite". (Page 15 line 1)

The expression "...suggest the importance to consider..." has been modified to "...suggest the importance on considering ...". (Page 15 line 4)

The expression "...were estimated respectively as..." has been changed to "...were estimated respectively to be". (Page 15 line 14)

The original expression "...measurement uncertainty. Comparison between  $f_{WSOM}$  and  $k_{WSM}$  in the humidification branch shows high correlation, as presented in Fig. 8b." has been modified to "...measurement uncertainty, which is supported by the high correlation between  $f_{WSOM}$  and  $\kappa_{WSM}$  in the humidification branch (Fig. 8b)." (Page 16 lines 3–4)

**8. Lines 3&18 on p15, using "as" here is not right, please rephrase the sentences. Line 30, "causal"?**

Because we think the usage of "as" is correct, the text has not been changed. (Page 16 lines 7 and 22)

The original expression "Further investigations of causal relations between neutralization and  $\kappa_{WSM}$  (or  $\kappa_{PM0.95}$ ) in the humidification mode are required." has been modified to "Further investigations on the influence of the degree of neutralization of inorganic salts on  $\kappa_{WSM}$  (or  $\kappa_{PM0.95}$ ) in the humidification mode are required." (Page 17 lines 1–2)

**9. Line 8 on p16, please rephrase "another important point is that...."**

The expression "Another important point is that results from offline analyses..." has been modified to "Moreover, results from the offline analyses...". (Page 17 line 12)

**Responses to Anonymous Referee #3**

The manuscript by Deng et al. entitled reports the detailed comparison of online and offline analysis for submicron particles collected at Okinawa island, Japan. By comparing the online and offline analysis data, they demonstrated that the offline analysis of aerosol samples by the AMS can quantitatively be conducted. It is a good demonstration about the usefulness of offline AMS analysis. The research was carefully conducted. The manuscript is well organized. I suggest publication of the manuscript after addressing the following comments.

**Chemical characteristics of OA**

The manuscript compares mass concentrations of chemical species, especially focusing on compounds that are measurable by the AMS. I wonder if the mass spectra of the organic material for online and offline agree each other. The authors compare O:C ratios in Figure 3. It will also be useful if H:C ratios are provided.

The comparisons between offline and online analyses based on mass spectra of organics have been added as Text S8, which is referred in the main text as:

"Further comparisons between offline and online analyses based on mass spectra of organics are presented in Text S8." (Page 11 lines 10–11)

**Predictions hygroscopic properties**

The authors employed E-AIM for predicting hygroscopic properties of particles, including phase transition phenomenon. Influence of organic compounds on hygroscopic growth is discussed in the main text. It will be good if the potential influence of organic compounds on deliquescence/efflorescence phenomena were also to be discussed.

The influence of organic compounds on the phenomena of deliquescence/efflorescence has been discussed briefly in the second paragraph of Sect. 3.2. We do not go into more detail about this point because the uncertainty of DRH and ERH was not expected to be small, as suggested from the result of AS particles (Text S3).

**Interpretation of hygroscopicity**

In Figure 8, the authors compare the values of kappa with the mass fractions of wsom and ammonium-sulfate ratio. However, the data are scattered, especially at higher RH, suggesting that some other factors might also be influencing hygroscopicity. It would be better if the authors can provide some ideas on it. Although  $f_{WSOM}$  and  $R_{A/S}$  are considered as key characteristics of chemical composition that govern  $\kappa_{WSM}$ , they do not fully represent the variations of chemical composition. Presence of minor inorganic ions and a possible variation of the  $\kappa_{WSOM}$  may also contribute to  $\kappa_{WSM}$  values. In addition, the measurement uncertainty of the composition and the uncertainty of surface tension may also be responsible for the obtained relationship of  $\kappa_{WSM}$  with  $f_{WSOM}$  or  $R_{A/S}$ . Moreover, if the contributions of WSOM and WSIM to the water uptake were not fully additive (i.e., ZSR relationship does not fully holds), it could also affect the relationship. Although these points are not assessed in this study, they are now addressed in the revised manuscript.

"Although the variations of  $\kappa_{WSM}$  and  $\kappa_{PM0.95}$  may also be contributed by other factors (e.g., presence of minor inorganic ions, possible variation of  $\kappa_{WSOM}$ , and non-additivity of the contributions of WSOM and WSIM to the water uptake), they are not assessed here." (Page 15 lines 27–29)

We, however, interpret the results that the negative correlations between  $f_{WSOM}$  and  $\kappa_{WSM}$  were observed even though  $f_{WSOM}$  was in a narrow range of 0.16–0.31, which indicates the importance of the relative contributions of WSOM and WSIM (mainly sulfate + ammonium) to the hygroscopicity of WSM even in the case of sulfate rich aerosol. It should be noted that our assessment of repeatability (Text S4) suggests that the measurement uncertainty for  $g_f$  should be small.

**Other minor changes**

- 1) The quantification of WSOM and WISOM based on mass spectra (Text S5) has been redone by omitting the contribution of m/z 38. Related results have been updated.
- 2) Page 4 line 26: information on the area of the backup filter was added.
- 3) Page 5 line 3, the "3 g" has been corrected to "~3 g".
- Page 5 line 9: "For TOC analyses" has been changed to "For total organic carbon (TOC) analyses".

- 5) Page 5 line 11: "for IC analyses" has been changed to "for ion chromatograph (IC) analyses".
- Page 5 line 20: "...(NH1, MH-110-12F-4; Perma Pure LLC),..." has been changed to "...(NH1, 94–98 % RH; MH-110-12F-4; Perma Pure LLC),...".
- 7) The original Table S4 is now Table S5, and the original Table S5 is now Table S6.
- 8) Page 12 line 21, "(57–59%)" has been changed to "(57–59%)".
- Page 14 line 16, "...aerosol liquid water mass exists." has been modified to "...aerosol liquid water mass should exist.".
- 10) Page 15 line 26, "Here, the influences of WSOM and..." has been modified to "Here, the influences of the mass fraction of WSOM (*f*WSOM) and...".
- 11) Page 15 line 31, "The relation between the mass fraction of WSOM ( $f_{WSOM}$ ) and  $\kappa_{WSM}$  at..." has been changed to "The relation between  $f_{WSOM}$  and  $\kappa_{WSM}$  at...".
- 12) Page 16 line 18, "Therefore, the deliquescence of ammoniated sulfate..." has been changed to "Therefore, the efflorescence of ammoniated sulfate...".
- 13) Page 21, the reference of Boreddy et al., 2018 has been deleted.
- 14) Page 26 lines 8–9, the reference of Tang, 1996 has been added.
- 15) In Text S3, "...E-AIM III model (Text S5, Fig. S2a)." has been changed to "...E-AIM III model (Text S6, Fig. S2a)."
- 16) Fig. S15 and its caption have been updated to indicate that  $\kappa_{WSOM}$  and  $\kappa_{EOM}$  are data in the dehumidification branch.
- 17) The caption of Fig. S20 has been corrected.

- 18) In text S1, the sentence "Note that, for both V- and W-modes, effective ambient measurement data were not collected temporarily also in other times during the campaign." has been added.
- 19) The labels of the right axes in Fig. 5 have been corrected.
- 20) The part "The mean number/volume concentrations of atmospheric aerosols are also shown." in the caption of Fig. 5 has been modified to "The mean number/volume-size distributions of atmospheric aerosols are also shown. The size distribution and  $\kappa_{online}$  of the aerosols at the study site are also discussed in Cai et al. (2017).
- 21) The sentence "The composition and size distribution of the aerosols at the study site are also discussed in Cai et al. (2017)." has been added to the end of the caption of Fig. 2.

(For details in the changes made, please see the track-change-mode version of the manuscript.)

**References**

[revised manuscript text omitted]